# Estimating lockdown-induced European NO₂ changes using satellite and surface observations and air quality models.

Jérôme Barré[1], Hervé Petetin[2], Augustin Colette[3], Marc Guevara[2], Vincent-Henri Peuch[1], Laurence Rouil[3], Richard Engelen[1], Antje Inness[1], Johannes Flemming[1], Carlos Pérez García-Pando[2,4], Dene Bowaldo[2], Frederik Meleux[3], Camilla Geels[5], Jesper H. Christensen[5], Michael Gauss[6], Anna Benedictow[6], Svetlana Tsyro[6], Elmar Friese[7], Joanna Struzewska[8], Jacek W. Kaminski[8,9], John Douros[10], Renske Timmermans[11], Lennart Robertson[12], Mario Adani[13], Oriol Jorba[2], Mathieu Joly[14], Rostislav Kouznetsov[15]

[1] European Centre for Medium-Range Weather Forecasts (ECMWF), Shinfield Park, Reading, UK

[2] Barcelona Supercomputer Centre (BSC), Barcelona, Spain

[3] National Institute for Industrial Environment and Risks (INERIS), Verneuil-en-Halatte, France

[4] ICREA, Catalan Institution for Research and Advanced Studies, Barcelona, Spain

[5] Department of Environmental Science, Aarhus University, Roskilde, Denmark

[6] Norwegian Meteorological Institute, Oslo, Norway

[7] Rhenish Institute for Environmental Research at the University of Cologne, Cologne, Germany

[8] Institute of Environmental Protection - National Research Institute, Warsaw, Poland

[9] Institute of Geophysics, Polish Academy of Sciences, Warsaw, Poland

[10] Royal Netherlands Meteorological Institute (KNMI), De Bilt, the Netherlands

[11] Netherlands Organisation for Applied Scientific Research (TNO), Climate Air and Sustainability Unit, Utrecht, the Netherlands

[12] Swedish Meteorological and Hydrological Institute (SMHI), Norrköping, Sweden

[13] Italian National Agency for New Technologies, Energy and Sustainable Economic Development (ENEA), Bologna, Italy

[14] CNRM, Université de Toulouse, Météo-France, CNRS, Toulouse, France

[15] Finnish Meteorological Institute (FMI), Helsinki, Finland

*Correspondence to*: Jérôme Barré (jerome.barre@ecmwf.int)

**Abstract.** This study provides a comprehensive assessment of NO₂ changes across the main European urban areas induced by COVID-19 lockdowns using satellite retrievals from the Tropospheric Monitoring Instrument (TROPOMI) onboard the

Sentinel-5p satellite, surface site measurements, and simulations from the Copernicus Atmosphere Monitoring Service (CAMS) regional ensemble of air quality models. Some recent TROPOMI-based estimates of changes in atmospheric NO₂

concentrations have neglected the influence of weather variability between the reference and lockdown periods. Here we provide weather-normalised estimates based on a machine learning method (gradient boosting) along with an assessment of the biases that can be expected from methods that omit the influence of weather. We also compare the weather-normalised

satellite-estimated $NO_2$ column changes with weather-normalised surface $NO_2$ concentration changes and the CAMS regional ensemble, composed of 11 models, using recently published estimates of emission reductions induced by the lockdown. All estimates show similar $NO_2$ reductions. Locations where the lockdown measures were stricter show stronger reductions and, conversely, locations where softer measures were implemented show milder reductions in $NO_2$ pollution levels. Average reduction estimates based on either satellite observations (-23%), surface stations (-43%) or models (-32%) are presented,

showing the importance of vertical sampling but also the horizontal representativeness. Surface station estimates are significantly changed when sampled to the TROPOMI overpasses (-37%) pointing out the importance of the variability in time of such estimates. Observation-based machine learning estimates show a stronger temporal variability than model-based estimates.

## 1.    Introduction

Nitrogen dioxide (($NO_2$; together with NO, a constituent of $NO_x=NO+NO_2$) is a very well-established cause of poor air quality in the most urbanized and industrialized areas of the world. $NO_2$ is harmful for living organisms by long-term atmospheric concentration exposure. It also plays a major role in urban ozone formation and secondary aerosols which are also harmful for living organisms at high levels in the lower atmosphere (Lelieveld et al., 2015; Myhre et al., 2013). According to the European Environment Agency (EEA, 2020a) the main European anthropogenic $NO_x$ sources are road transport (39%),

energy production and distribution (16%), commercial, residential and households (14%), energy use in industry (12%), agriculture (8%), non-road transport (8%) and industrial processes and product use (3%). With an atmospheric lifetime typically below 1 day, $NO_x$ is relatively short-lived and is mainly controlled by photochemical reactions. The majority of $NO_x$ therefore does not get transported far downwind from its sources (Seinfeld and Pandis, 2006). Thus, near-surface $NO_x$ concentrations are high over cities and densely populated areas and low otherwise. Besides emissions, the variability of $NO_x$

is strongly driven by meteorological conditions, especially atmospheric transport, vertical mixing, and solar radiation, affecting the level of accumulation close to the emission sources (Arya, 1999). For example, increased wind speed and a higher planetary boundary layer height will increase the dispersion of $NO_x$ from the emission sources. It is this short lifetime, which is partly modulated by atmospheric conditions such as temperature and radiation combined with localized emission sources, that make $NO_2$ an excellent proxy for detecting emission reductions, from both surface and satellite measurements.

The worldwide outbreak of the coronavirus disease (COVID-19), which arose in late 2019 in China and spread around the world in early 2020, led many countries to take action to slow down the infection growth rate of the virus. The so-called lockdowns severely restricted or banned movements of people closed most public places and limited journeys to essential work commutes. Some measures started in China in late 2019 with stricter lockdowns in January 2020. In Europe, lockdown

measures were implemented at various dates during February and March 2020. These lockdowns drastically reduced traffic and also activity levels in most industries (Guevara et al., 2020; Le Quéré et al., 2020). These sectors represent a large share of $NO_x$ emissions (51% according to EEA 2020a). Studying $NO_2$ concentration changes during the lockdown is therefore very important to assess the impact of such activity-level reductions on a population's exposure to pollution. The COVID-19 lockdown is a unique opportunity to assess the impact of future pollution reduction measures, in particular, the impact of drastic reductions on the road transport sector using combustion energy.

The lockdowns were expected to have large effects on urban $NO_2$ air pollution levels in conjunction with other modulating factors (i.e., weather conditions). The first quarter of 2020 had specific and highly variable meteorological conditions. Storm Ciara crossed over Europe in the second week of February followed by Storm Dennis that crossed Europe a week later. Both extratropical storms generated strong winds over the northern half of Europe (above 45°N) from February 9th, 2020 until February 18th, 2020. Strong winds, yet milder than during storms Ciara and Dennis, were also generated by storms Karine and Myriam over the Iberian Peninsula, the southern part of France and the northern part of Italy in the first week of March. Moreover, February and March 2020 displayed stronger positive temperatures anomalies over Europe in comparison with February and March 2019 (https://surfobs.climate.copernicus.eu/stateoftheclimate). Such weather anomalies, however, did not persist during the second quarter of 2020. Accounting for the effect of such meteorological variations is very important to assess accurately the effect of COVID-19 related mobility restrictions on air pollution. Different approaches can be used to assess the pollution changes, based on different types of data, such as satellite observations, surface site observations and air quality models.

Several studies used the recently launched (October 2017) Tropospheric Monitoring Instrument (TROPOMI, Veefkind et al., 2012) onboard the Copernicus Sentinel-5 Precursor (S5P) satellite to highlight the $NO_2$ reductions caused by the COVID-19 lockdowns. The substantial interannual variability of meteorological conditions together with the young age of the instrument prevented estimating a representative climatological baseline to which $NO_2$ levels observed during the lockdown period could be compared. As a result, satellite-based studies using TROPOMI comparing before and after lockdown periods (e.g., Wang et al., 2020b) or comparing the lockdown period with its 2019 equivalent (e.g., Bauwens et al., 2020, Nakada et al., 2020, Zambrano-Monserrate et al. (2020)) have given little to no weight to the synoptic meteorological conditions and how they could potentially flaw the emission change estimates.

In contrast, Schiermeier (2020) mentioned the 'weather factor' early on in the COVID-19 crisis, which can strongly affect the pollution levels. And studies, such as Le et al. (2020), showed 2019 and 2020 TROPOMI $NO_2$ comparisons but acknowledged the impact of weather anomalies on pollution levels. It is only very recently that a weather-normalisation technique has been applied to estimate $NO_2$ changes due to the COVID-19 restrictions across cities in the US based on TROPOMI (Goldberg et al., 2020). Yet, such analyses place insufficient importance and provide insufficient clarity about the fact that satellite data used in such analyses are conditioned by the cloud coverage, revisit frequency and quality flag of the satellite observations. Ignoring or not acknowledging such information can also lead to flawed satellite-based estimates and

provide misleading information (https://atmosphere.copernicus.eu/flawed-estimates-effects-lockdown-measures-air-quality-derived-satellite-observations).

Several studies have investigated lockdown impacts using surface measurement sites. For example, Wang et al. (2020a) showed that lower emissions from motor vehicles and secondary industries were most likely responsible for the observed decreases of $NO_2$ concentrations in China during January-March 2020. Collivignarelli et al. (2020) showed using surface station measurements that major $NO_2$ reductions occurred in Milan, a city that showed a rapid increase of cases early in the European COVID-19 crisis (February 2020) and was one of the first cities to be put into lockdown in Europe. Past studies such as Carslaw and Taylor (2009) showed the usefulness and the importance of weather normalisation techniques for air pollution applications based on surface observations, such as the local air traffic activity impact on $NO_2$ predictions. This was followed more recently by Grange et al. (2018, 2019) where machine learning techniques were used to perform weather normalisation for analysing trends and detecting the impact of policy measures on air quality. Built on this previous work, several studies made use of machine learning to estimate the impact of the COVID-19-related mobility restrictions on air pollution levels, taking into account the confounding effect of the meteorological variability. Using ML models fed with ERA5 reanalysis meteorological data, Petetin et al. (2020) highlighted a strong reduction of surface $NO_2$ concentrations across most Spanish urban areas during the first weeks of lockdown. Similarly, Keller et al., 2020 assessed the $NO_2$ pollution changes using worldwide surface measurements showing country-dependent variations on reductions.

Finally, air quality modelling systems offer a valuable tool for representing the evolution of pollutants in the atmosphere according to changes in emissions, physical processes and weather variability. The Copernicus Atmosphere Monitoring Service (CAMS) produces daily European air quality forecasts and analyses using an ensemble of 11 models ensuring unique reliability and quality (Marecal et al., 2015). Using emission scaling factors to account for lockdown measures such an ensemble of models can be used to estimate lockdown reductions on $NO_2$ pollution (amongst other pollutants) and account for the weather variability at the same time (Colette et al., 2020, Guevara et al., 2020).

This paper aims at providing a comprehensive and comparative assessment of the impact of the first European COVID-19 lockdown on $NO_2$ pollution levels over major European urban areas using satellite observations, surface in-situ observations, and air quality models. We firstly illustrate how misleading it can be to ignore the influence of the weather variability when assessing the lockdown-induced changes of $NO_2$ with TROPOMI. Then, in order to quantify these changes, we use ML-based weather-normalisation methods for estimating the "business as usual" (BAU) $NO_2$ pollution levels that would have been observed without any lockdown measures, based on both TROPOMI $NO_2$ tropospheric columns (Section 2) and surface in-situ observations (Section 3). $NO_2$ changes are then investigated with the CAMS regional ensemble (Section 4). We compare and discuss the three different approaches in Section 5 followed by conclusions in Section 6.

## 2. TROPOMI NO₂ column estimates

### 2.1. Dataset and analysis periods

We use the operational Copernicus S5P TROPOMI $NO_2$ level-2 product, for which data have been available since 28 June 2018. These observations are tropospheric columns (from the surface to the top of the troposphere) with a pixel resolution of 5.5km by 3.5km since 6 August 2019 and 7km by 3.5km before. The instrument can have an up-to-daily revisit at 13:30 mean local solar time assuming clear-sky conditions. The quality flag (qa) provided with the retrieval is used to select only good quality data (qa > 0.75), which removes cloud-covered scenes, errors, and problematic retrievals (Eskes et al., 2019). The TROPOMI data are then binned on a regular $0.1° \times 0.1°$ grid to perform statistical analyses and to facilitate the processing of time series for the locations of interest, i.e., large European cities in this study (see section 2.2), as well as the comparison with other datasets such as the $0.1° \times 0.1°$ CAMS regional air quality models (Marecal et al., 2015) and the 9 km resolution weather forecasts from the European Centre for Medium-Range Weather Forecasts (ECMWF).

In this study we consider February, March and April 2020 and 2019 to assess the changes in $NO_2$ columns due to COVID-19 restrictions over Europe. Although the lockdown conditions and dates vary between countries, we consider the 15[th] of March 2020 as a representative starting date for the European-wide lockdown, given that most European countries implemented their nation-wide social distancing measures along the 2-week period from 9 March 2020 (Italy) to 23 March 2020 (United Kingdom (UK)). Two periods of the year are considered in this study: the pre-lockdown period from 1 February to 15 March, and the lockdown period from 16 March to 31 April. This study thus focuses on the most stringent period of the first European lockdown (since many countries then started to ease up their lockdown restrictions from the beginning of May onwards).

In Figure 1, mean TROPOMI $NO_2$ tropospheric columns are displayed for the pre-lockdown and lockdown periods in 2020 and their equivalents in 2019. The comparison of pre-lockdown and lockdown averages for 2020 only shows a decrease in Southern Europe but no clear reduction at more northern latitudes (i.e., the UK, The Netherlands and Germany). In the corresponding 2019 pre-lockdown period much larger $NO_2$ columns are seen than in 2020. During this period of the year, the meteorological conditions over Northern Europe were significantly different between 2019 and 2020. A number of named extratropical cyclones (storms Ciara, Denis, Karine and Myriam), combined with a strong positive anomaly in surface temperature, occurred over Europe during February and early March 2020, especially in western and northern Europe. Such anomalies in wind and temperature were not observed in 2019. Figure 2 shows the distribution of 10-meter wind speed, planetary boundary layer (PBL) height and 2-meter temperature from the 9 km operational forecasts from the ECMWF Integrated Forecasting System (IFS) in both 2019 and 2020 for the pre-lockdown and lockdown periods at the S5P overpass times. Details on how the PBL height is calculated can be found in the IFS documentation (part IV, chapter 3 in https://www.ecmwf.int/en/elibrary/19748-part-iv-physical-processes). Before 15 March, these parameters show very different distributions with much lower values in 2019 than in 2020, i.e., less circulation and less vertical diffusion under colder conditions. These differences in meteorological conditions explain the increase of $NO_2$ tropospheric columns in 2019 compared

160 to 2020. Conversely, during the post-15 March period, the meteorological distributions are more similar showing much smaller differences. This illustrates the need for accounting for the meteorological effect when assessing the changes of $NO_2$ tropospheric columns associated with the lockdown.

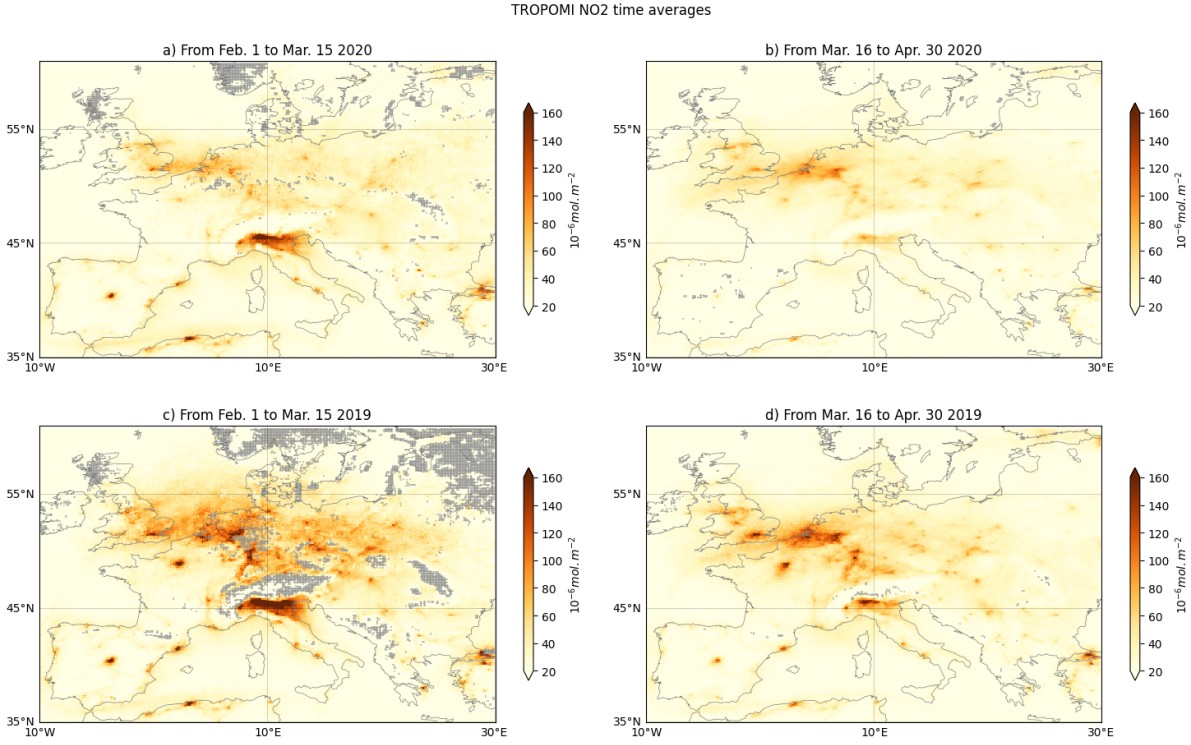

**Figure 1. Average maps of the TROPOMI $NO_2$ tropospheric columns (mol.m$^{-2}$) for European pre-lockdown and lockdown periods in 2020 (a, b respectively) and corresponding periods in 2019 (c, d). Grey areas indicate where the number of revisits is strictly below 5.**

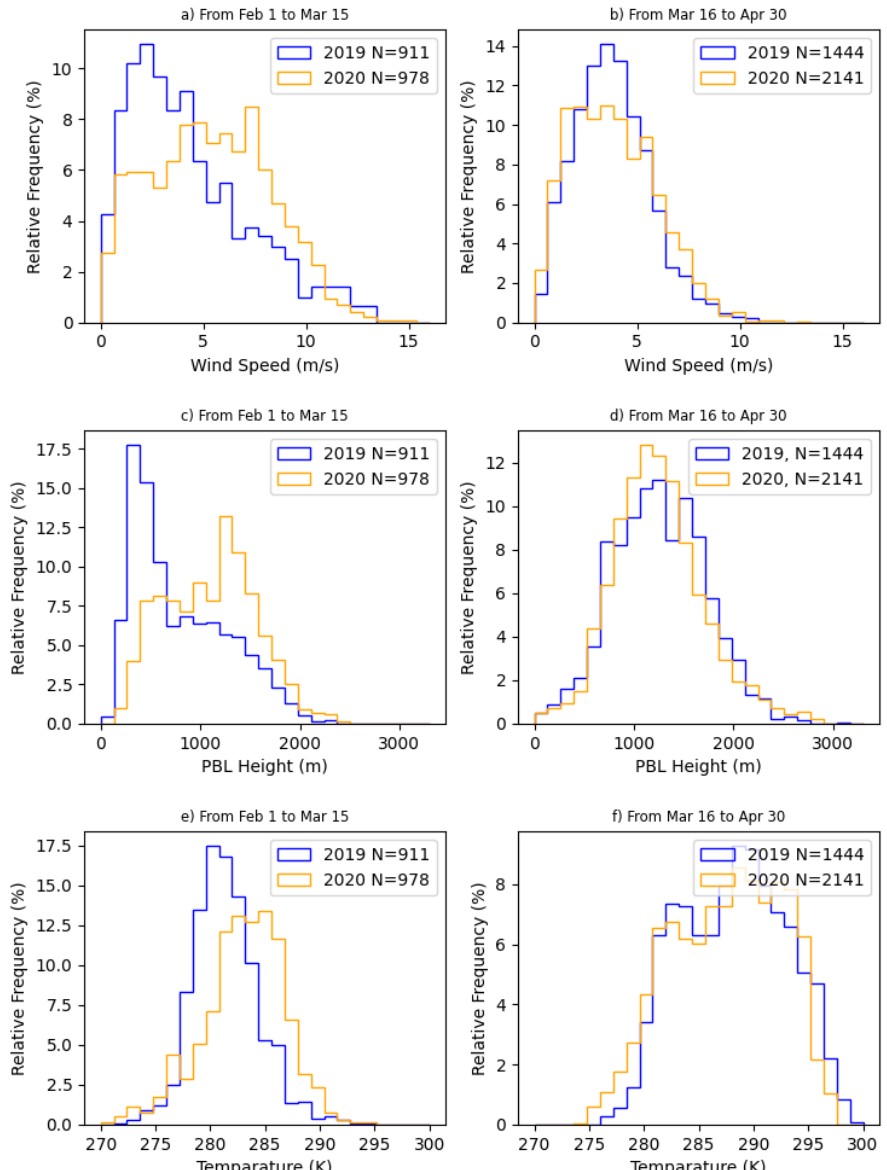

Figure 2. Probability density functions of 10-meter wind speed (m.s$^{-1}$, first row), planetary boundary layer (PBL) height (m, second row), and 2-meter temperature (K, third row) from the ECMWF operational forecasts for European periods before (a,b) and after 15 March (c,d), comparing 2020 to 2019. Distribution is computed for urban areas above 0.5 Million inhabitants between 10°W,20°E,45°N and 60°N at the S5P overpasses times. N is the sample size for each distribution that can be multiplied by the relative frequency (in %) to obtain the absolute frequency.

## 2.2. Non-weather-normalised changes of TROPOMI $NO_2$ tropospheric columns

Changes in $NO_2$ tropospheric columns associated with the lockdown measures can be estimated by comparing $NO_2$ levels observed during the lockdown period in 2020 with a given baseline. In this section, we compare the results obtained with two different baselines: (1) the $NO_2$ levels observed during the pre-lockdown period in 2020 (hereafter referred to as the "before-during" approach), (2) the $NO_2$ levels observed during the same period of the year in 2019 (hereafter referred to as the "year-to-year" approach). We focus our study on the largest European urban areas for which the city population is exceeding 0.5 million inhabitants (according to the population database provided by https://simplemaps.com/data/world-cities), resulting in a total of 100 locations. Assessing the changes of $NO_2$ tropospheric columns from satellite observations is more challenging over rural areas as the $NO_2$ levels are much lower than over urban areas. Because of the much lower $NO_2$ tropospheric column values over rural areas, the relative estimates of pollution reduction are very sensitive to small changes in the tropospheric columns and therefore also to instrument noise. We choose the observations with footprints closest to the European city centres and with more than 5 data points per pre-lockdown and lockdown period. If this condition is not reached, the location is discarded from the analysis. The "before-during" estimate corresponds to the difference between the pre-lockdown and the lockdown period median estimates. Figure 3 shows changes calculated for 2020 (Fig. 3b) and the equivalent for 2019 (Fig 3a) for comparison. This method shows drastic $NO_2$ reductions by more than 75% in 2020 for most of the southern European large urban areas. Reductions are, however, not obvious over northern European urban areas and show strong variations from one location to another. For example, over the UK and Belgium, some urban areas show increases well above 30%, while other urban areas show reductions even though the same lockdown measures were applied nationwide. Applying the same method over 2019, a similar strong decrease of $NO_2$ levels over many major European urban areas is visible. Such reductions in 2019 are not expected in relation to COVID-19 lockdown measures. Therefore, such "before-during" type of satellite-based estimates do not provide a robust methodology for assessing the effects of COVID-19 lockdown on European $NO_2$ pollution levels.

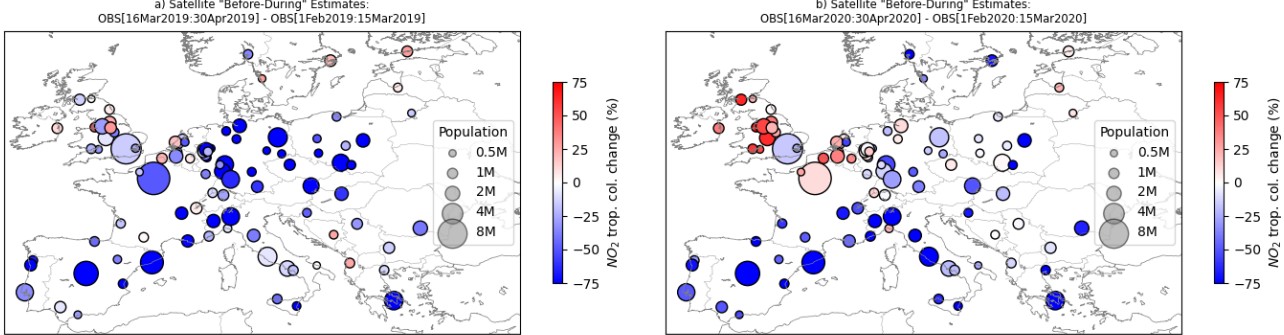

**Figure 3. "Before-during" estimates of TROPOMI tropospheric NO₂ column change (%) for urban areas above 0.5**

**million inhabitants, in 2019 (a) and 2020 (b). The diameter of the circles is proportional to the population count in**

**each urban area.**

The "year-to-year" approach has been more widely used in scientific publications and web news articles and consists of comparing observations from 2020 to observations from 2019 over the period of interest. Figure 4 shows such "year-to-year" estimates, comparing the median values between 2020 and 2019, for the pre-lockdown (Fig. 4a) and lockdown (Fig. 4b)

periods. During the lockdown, an overall reduction is seen all over Europe with more moderate reductions over southern Europe compared to the "before-during" estimates (see Fig. 3b). Changes over northern Europe do not show strong variations between the various urban areas as was visible in the "before-during" method. An overall decrease is seen over most European locations, with the strongest reductions in European countries (e.g., France, Spain or Italy), where lockdown measures were more stringent (according to the Oxford Coronavirus Government Response Tracker stringency index Hale et al., (2020)).

However, looking at the pre-lockdown estimates, northern Europe also shows drastic negative changes, that are larger than during the lockdown period. Such changes in pollution levels across Europe should not be expected if only the impact of emission changes was considered. The "year to year" method thus appears to be strongly dependent on the interannual NO₂ variability, where meteorology plays a crucial role. Although it respects the seasonality on NO₂, this method could still lead to large errors when assessing differences in NO₂ levels and more generally the pollution level reductions due to the COVID-

19 lockdown.

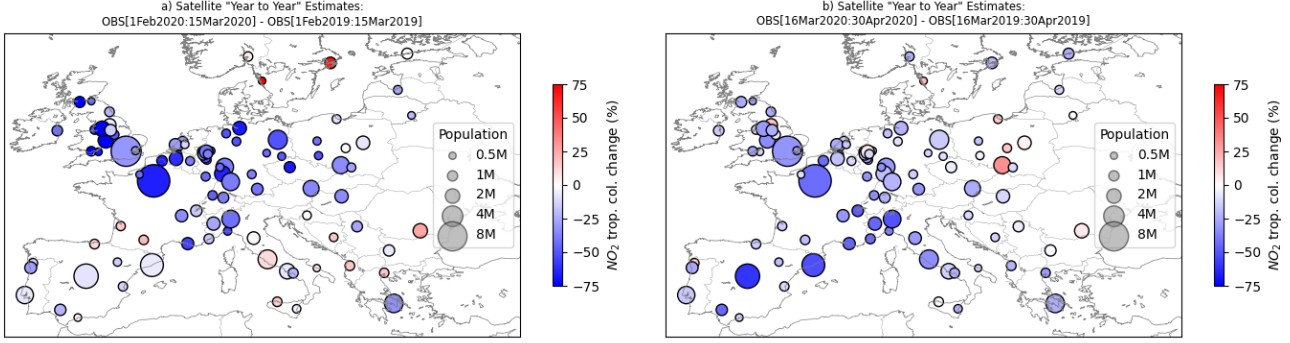

**Figure 4. "Year-to-year" estimates of TROPOMI tropospheric NO₂ column change (%) for urban areas above 0.5**

**million inhabitants, in 2019 (a) and 2020 (b). The diameter of the circles is proportional to the population count in**

**each urban area.**

## 2.3. Weather-normalised changes of TROPOMI NO₂ tropospheric columns

### 2.3.1. Methods

Weather-normalisation methods account for weather variability to more accurately estimate the net changes of NO$_2$ induced by the lockdown in urban areas. Previous studies have used meteorological and air pollution predictors to build simplified models for the simulation of satellite observations or to generate predictions of atmospheric composition (e.g., Worden et al., 2013, Barré et al., 2015). In this study, we use a novel approach for the simulation of TROPOMI satellite observations under BAU conditions, i.e., in the absence of lockdown restrictions, based on the Gradient Boosting Machine (GBM, Friedman, 2001) regressor technique. GBM is a popular decision tree-based ensemble method belonging to the boosting family. For the predictors, we use the following weather and air quality variables from the ECMWF and CAMS operational forecasts at 9 km and 0.1° resolutions, respectively: 10-m wind speed and direction, PBL height, 2-m temperature, surface relative humidity, geopotential at 500hPa, and NO$_2$ surface concentrations from the CAMS regional ensemble forecasts. The NO$_2$ surface concentrations used here are obtained from the CAMS operational regional forecasts, which are based on business-as-usual emission information and are therefore different from the simulations presented in section 4. In the CAMS regional forecast product, there is also no assimilation of observations to constrain the forecasts. Therefore, the NO$_2$ surface concentrations used to train and make model predictions do not include lockdown effects and are independent of the air quality model pollution change estimates provided in section 4. Additionally, the following time and location variables were also included in the set of predictors: latitude, longitude, population, Julian date (number of days since January 1$^{st}$) and weekday (to reflect expected weekend/weekday effects). A quite similar machine learning (ML)-based approaches have already been successfully applied to in-situ surface AQ observations (e.g., Grange et al., 2018, 2019, Petetin et al., 2020). We use data from 1 January 2019 to 31 May 2019 as a training set and apply the model to 2020 to generate simulations of BAU NO$_2$ tropospheric columns. For validation purposes, we have randomly split the input data in a 90% and 10% share for training and testing, respectively. Hyperparameter tuning (see annex A for details) was performed using a grid search method with 5-fold cross-validation and using the ranges indicated by Petetin et al. (2020). In contrast to Petetin et al. (2020), who trained one ML model per surface air quality monitoring station, only one single ML model is trained here for all cities. This choice is motivated by the small size of the available training dataset (about 10,000 data points, see Table 1). After the hyperparameter tuning and evaluation of the model, the BAU observation simulations have been generated using 100% of the January-May 2019 dataset to use the maximum amount of data points possible.

| | MB [$10^{-6}$mol.m$^{-2}$] | nMB [%] | RMSE [$10^{-6}$mol.m$^{-2}$] | nRMSE [%] | PCC | N |
|---|---|---|---|---|---|---|
| S5P training set | 0.00 | +0.02 | 1.4 | 45.68 | 0.87 | 9,634 |

| S5P | -0.04 | -1.30 | 1.68 | 56.38 | 0.79 | 1,071 |
| test set | | | | | | |

**Table 1. Performance of the machine learning simulations of NO₂ tropospheric columns over all European urban areas included in the dataset. The training set and testing set cover January-May 2019 and are randomly sampled (90% and 10%, respectively) over that period. Shown are the mean bias (MB), normalised mean bias (nMB), root-mean-square error (RMSE), normalised root-mean-square error (nRMSE), Pearson correlation coefficient (PCC) and the number of data points (N).**

### 2.3.2. Results

Detailed scores of the performance of the gradient-boosting regressor with respect to TROPOMI observations, such as mean bias (MB), normalised mean bias (nMB), root-mean-square error (RMSE), normalised root-mean-square error (nRMSE) and the Pearson Correlation Coefficient (PCC), can be found in Table 1. In order to check for obvious cases of overfitting (i.e., when the GBM model is fitting the data used for training too closely and is thus lacking generalization skills regarding new data), results are shown for both training and testing datasets. The statistics for the training set and the testing set show similar results, such as low bias, good correlation, but significant RMSE values. The statistical performance obtained for the training set indicates that there is no clear sign of overfitting in the predictions. Since TROPOMI data are only available from mid-2018 onwards, the training set is relatively small. For this reason, the predictions are featuring significant RMSE values and will have a large random error. The RMSE values stay, however, within a similar range as for the surface site air quality ML predictions, as shown in Section 3 and Table 2. The low mean bias and high correlation values indicate that the main BAU NO₂ tropospheric column variability is represented without large systematic errors. Subtracting the BAU NO₂ simulated columns from the actual observed NO₂ columns during the lockdown period (from 16 March 2020 to 30 April 2020) gives us an estimate of the reductions in the NO₂ background levels over the urban areas considered in this study. Figure 5 provides an example of a time series over Madrid that shows the behaviour of the GBM against the real observations for 2019 (the training period) and 2020 (the actual simulation period). In 2019, the GBM predictions follow the variations seen in the observations but do, however, also show differences, either being above or below the observations. In 2020, similar behaviour is observed until the lockdown date where the GBM predictions show consistently higher values than the observations, but still, follow the same variations as the observations. This shows that the GBM predictions based on BAU predictors are performing realistically and account for the variability in the BAU scenario. This therefore, provides a method to assess the pollution changes due to lockdown restrictions using satellite data more robustly than the "before-during" or "year-to-year" methods.

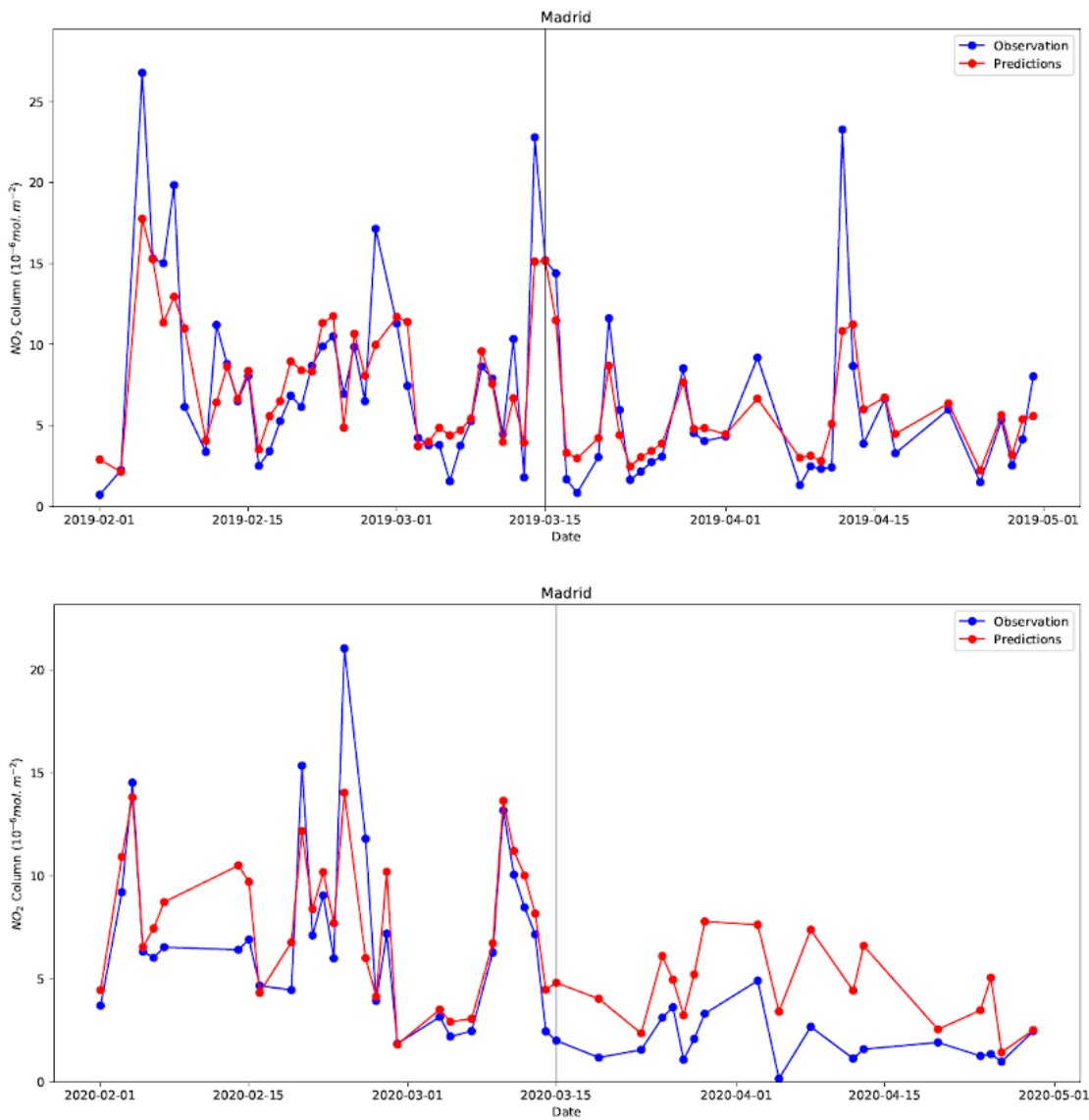

**Figure 5. Example of a time series over Madrid illustrating the performance of the machine learning NO₂ column predictions for February-March-April 2019 (top panel) and the same period in 2020 (bottom panel).**

Figure 6 shows the equivalent estimates as in Fig. 3 and 4 for the pre-lockdown and lockdown periods using the ML-based BAU estimates as the baseline. The estimates of the $NO_2$ changes are based on the median value of the real observation minus the simulated BAU observation distributions. As shown in Table 1, the GBM performance shows large RMSE values which can sometimes result in significant outliers due to the small training set used. We choose to display the median to avoid the influence of potential outliers in the estimates as much as possible. The pre-lockdown ML-based estimates do not show as

strong of an overall reduction as in the "year-to-year" (Fig. 4) or "before-during" (Fig. 3) estimates. A summary of the average and the standard deviation of the set of median estimates across all the considered European urban areas is provided in Table 2 for each of the satellite methods. While both "year-to-year" and "before-during" methods showed substantial changes (24% and 30% respectively) of $NO_2$ during the periods outside lockdown (i.e., in 2019 or before the lockdown in 2020) when low to no reduction should be expected, the ML-based weather-normalisation method provides changes closer to 0%, which are

considered to be more realistic.

       The weather-normalisation method is not devoid of uncertainties and can, in particular, be affected by trends in $NO_2$ levels. With a known trend seen in European $NO_x$ emissions of around 2 to 4% per year (EEA, 2020a) and only one year to train the data, the ML method potentially provides a stronger than expected overall reduction of around 8%. The "before-during" and the "year-to-year" approaches also show stronger reduction estimates on average during 2019 and the pre-

lockdown period, respectively. The latter two methods also display a stronger standard deviation across cities than the weather-normalisation method, which suggests substantial local biases due to the omission of the meteorological variability.

       When we consider the lockdown period, the weather parameter distributions are much more similar between 2019 and 2020 (Figure 2) than is the case for the pre-lockdown period, and on average, across Europe, the "year-to-year" and weather-normalised estimates show results within the same range in terms of mean (around -20%) and variability amongst the

median estimates obtained for all urban areas (around 16%). This is, however, not the case for the "before-during" estimates, which show much stronger variability between European urban areas (62%). The "before-during" estimates are therefore not reliable and the "year-to-year" method is very dependent on the differences in the meteorological situations between 2019 and 2020. For this reason, the ML estimates are the most reliable and will be used solely for the rest of this study. Details of the ML estimates during the lockdown provided in Fig 6 are reported in the table in annex B. The $NO_2$ tropospheric column change

estimates (median values per urban area) show on average a reduction of 23%, but urban areas that are known to have the most stringent measures (Hale et al., 2020) show much stronger reductions, e.g., Madrid 60%, Barcelona 59%, Turin 54%, and Milan 49%. Lighter reductions can be observed in urban areas where less stringent measures were taken, e.g., Stockholm 17%. To check the robustness of these results, equivalent estimates are provided using surface stations and air quality models in section 3 and 4 and will be compared in section 5.

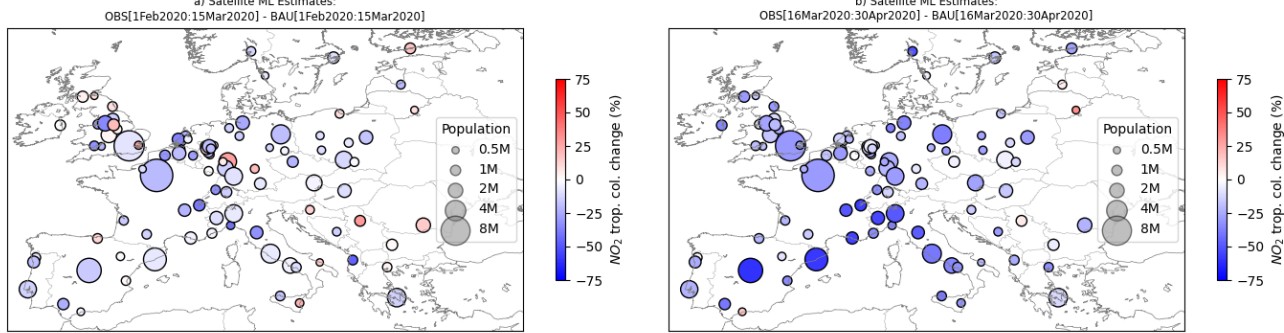

**Figure 6. TROPOMI-based estimation of tropospheric NO₂ column change (%, relative to the BAU predictions) for urban areas with at least 0.5 million inhabitants computed using the ML-based weather-normalisation method for the pre-lockdown and lockdown periods (a and b respectively). The diameter of the circles is proportional to the population count in each urban area.**

| | Mean (%) | Standard Deviation (%) |
|---|---|---|
| "Before-during" [2019] | -40 | 47 |
| "Before-during" [2020] | -25 | 62 |
| "Year-to-year" [01/02 to 15/03] | -26 | 31 |
| "Year-to-year" [16/03 to 30/04] | -18 | 16 |
| Machine Learning [01/02 to 15/03] | -8 | 16 |
| Machine Learning [16/03 to 30/04] | -23 | 16 |

**Table 2. Scores over all European urban areas included in the dataset for the different TROPOMI NO₂ tropospheric columns change estimates. Mean and standard deviation are calculated for the median estimates of all urban areas considered in the study, i.e., the standard deviation is a metric of the inter-urban area spread.**

## 3. Surface station estimates

### 3.1. Methods

We have estimated the impact of the COVID-19 lockdown on surface NO₂ pollution in European areas using the methodology introduced by Petetin et al. (2020), applied to up-to-date (i.e., partly unvalidated real-time) hourly NO₂ data from the European Environmental Agency (EEA) AQ e-Reporting (EEA, 2020b). We first selected the urban/suburban background stations located within 0.1° from the city centres and applied the quality assurance and data availability screening described in

Petetin et al. (2020), using the GHOST metadata (Globally Harmonised Observational Surface Treatment, Bowdalo et al., 2020, in preparation). A total of 164 stations in 77 urban areas was selected. At each station (independently), we estimated the BAU $NO_2$ mixing ratios that would have been observed during the lockdown period under an unchanged emission forcing. This was done using GBM models fed with meteorological inputs (2-m temperature, minimum and maximum 2-m temperature, surface wind speed, normalised 10-m zonal and meridian wind speed components, surface pressure, total cloud cover, surface net solar radiation, surface solar radiation downwards, downward UV radiation at the surface and PBL height) taken from the 31 km horizontal resolution ERA5 reanalysis dataset (Hersbach et al., 2020) in addition to other time features (date index, Julian date, weekday, hour of the day). The ERA5 reanalysis data set is a consistent model version over time but at coarser resolution in comparison to the ECMWF high-resolution operational forecasts used in the TROPOMI estimates (31 km versus 9 km).

All GBM models were trained and tuned with data for the past 3 years (2017-2019) and tested with data from 2020 before the lockdown. Petetin et al. (2020) showed that such duration for training the GBM models is generally sufficient for capturing the influence of the weather variability on surface $NO_2$ mixing ratios. As discussed in more detail in Petetin et al. (2020), the date index feature here allows limiting the potential issues related to the presence of trends in the $NO_2$ time series (between 2% to 4% decrease per year, EEA 2020a). If a substantial trend exists, the GBM models will put more importance on this feature, which in practice will force the model to make $NO_2$ mixing ratio predictions (in 2020) in the range of the values observed during the last part of the training dataset, ignoring the oldest training data. Thus, given the long-term reduction of $NO_2$ resulting from policy measures across Europe, considering longer training periods is not expected to improve the performance of the GBM models. In contrast to Petetin et al. (2020), who predicted BAU $NO_2$ at a daily scale, the ML models developed here are predicting $NO_2$ at an hourly scale (in order to get results collocated in time with TROPOMI overpasses; see also below). We then deduced the weather-normalised $NO_2$ changes due to the lockdown by comparing observed and ML-based BAU $NO_2$ mixing ratios.

## 3.2. Results

Table 3 shows the overall performance of the GBM models on the training and test data sets. Statistical results are similar to the TROPOMI $NO_2$ GBM model. Biases are low and correlation is high and there is a significant RMSE. As explained in section 2.3.2, statistical scores in the training set and the test set suggest that there is no apparent sign of overfitting in the predictions showing reasonable performance. Note that the RMSE and PCC are deteriorated compared to the statistics obtained over Spain in Petetin et al. (2020), mainly due to the fact that we are here working with hourly estimates. This is demonstrated by similar results as those of Petetin et al. (2020) that are obtained over this set of European cities when predicting $NO_2$ at the daily scale (for the test dataset: nRMSE=28%, PCC=0.88, N=11,082).

| | MB [ppbv] | nMB [%] | RMSE [ppbv] | nRMSE [%] | PCC | N |
|---|---|---|---|---|---|---|
| Surface stations training set (2017-2019) | 0.0 | 0.0 | 5.53 | 40.88 | 0.84 | 4,048,696 |
| Surface stations test set (1 Jan 2020 – 15 Mar 2020) | +0.95 | +7.02 | 6.24 | 45.87 | 0.80 | 268,960 |

**Table 3. Performance of the ML predictions of hourly NO$_2$ surface mixing ratios over all European urban areas included in the dataset.**

For a stricter comparison with the results discussed in Section 2, we provide two different estimates to assess the satellite sampling effect: i) using all hourly values or ii) filtered according to the S5P satellite overpass time (13:30 local solar time) and 'qa' filtering (clear-sky only). Figure 7 displays relative change estimates, showing the median of the distributions for each European city above 0.5 million inhabitants. Overall, the estimates for both sampling strategies are broadly consistent,

with NO$_2$ reductions of around 37% and 43% on average for the hourly sampling and the S5P overpass sampling, respectively (Table 3). The surface station estimates also show geographical variations similar to the satellite estimates, with larger reductions corresponding to locations with more stringent lockdowns (i.e., Spain, Italy and France) and less stringent lockdowns (i.e., Sweden, Germany). For example, Madrid shows reductions of 61% and 60% using the hourly surface stations and the satellite overpass time sampled surface stations, which are very similar to the satellite estimates. In contrast, Stockholm

shows very small reductions of 8% and 3%, respectively. These latter values are different from the satellite-based estimates (reduction of 17%) and point out some uncertainty regarding the estimates in this area.

Northern Europe (particularly Germany, Poland and the UK) displays larger NO$_2$ reductions with the estimates at satellite overpass time. This points out a possible dependence on the time of the day in the emission and pollution reductions. In general, those NO$_2$ relative changes based on the surface in-situ observations are larger than the ones based on satellite NO$_2$

tropospheric columns. These two points are further discussed in section 5.

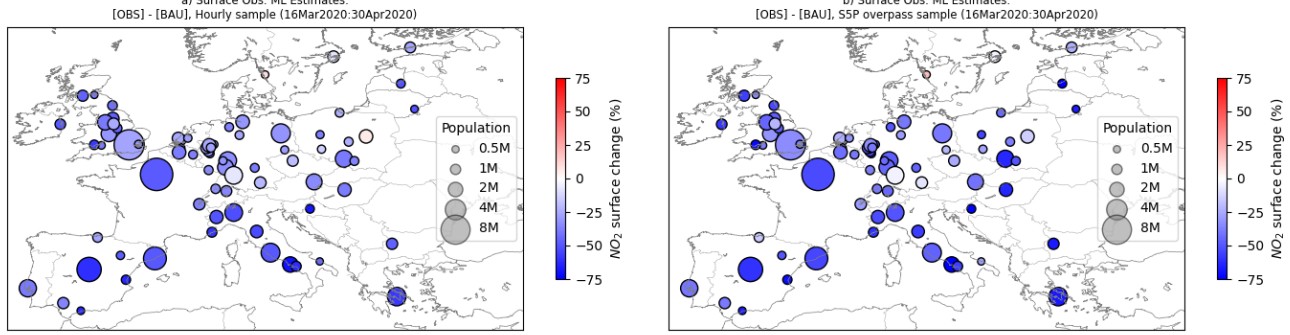

**Figure 7. Weather-normalised estimation of NO₂ changes (%, relative to the BAU predictions) using surface observations during the lockdown period using business as usual (BAU) simulated observations as the baseline for urban areas with at least 0.5 million inhabitants. The left-hand side (a) shows the estimates using full hourly datasets and the right-hand side (b) shows the estimates using the S5P overpass time sampled dataset. The diameter of the circles is proportional to the population in each urban area.**

## 4. CAMS regional ensemble model estimates

### 4.1. Methods

Model estimates have been calculated using the CAMS European regional air quality forecasting framework, which is an ensemble of 11 models (Marécal et al. 2015). These models are used to calculate multi-model median values, which is the best performing quantity on average compared to individual models. Using such a multi-model approach is useful to minimize the imperfections in each model formulation. Operational evaluation and validation of the CAMS European ensemble against independent observations is performed and delivered routinely and can be accessed at https://atmosphere.copernicus.eu/index.php/regional-services.

Two sets of model hindcasts have been conducted using two different emission scenarios: BAU emissions and reduced COVID-19 lockdown emissions. The emission inventory used for the BAU reference simulation is the same that is used in the daily Regional Air Quality Forecasts of CAMS for Europe, i.e., the CAMS-REG-AP dataset (v3.1 for the reference year 2016, Granier et al., 2019). It is compiled by TNO (Netherlands Organisation for Applied Scientific Research) under the CAMS emission service, based on official emissions reported by the countries to the EU (NEC Directive) and UNECE (LRTAP Convention /EMEP, Kuenen et al., 2014). The spatial resolution of the emissions is $0.1° \times 0.05°$ but re-gridded to $0.1° \times 0.1°$ to match the models' grid. The alternative emission scenario, corresponding to the lockdown period, was derived by combining the original CAMS-REG-AP inventory with a set of country- and sector-resolved reduction factors (Guevara et al., 2020). For

the present work, time-invariant emission reduction factors were used by country and for three activity sectors: manufacturing industry, road transport, and aviation (landing and take-off cycles) that are reduced on average by 15.5%, 54% and 94%, respectively. These sectors were considered to be the most affected by changes in activity during lockdown (Le Quéré et al., 2020).

The reduction factors were computed from collections of near-real-time activity data, such as Google Community Mobility Reports (Google LLC, 2020) for road transport, airport statistics from Flightradar24 (2020) for aviation and electricity load information from ENTSO-E (2020) for the industry sector. Results from Guevara et al., (2020) showed that during the most severe lockdown period (23 March to 26 April), estimated surface emission reductions at the European level were most important for $NO_x$ (33%) with road transport being the main contributor to total reductions in all cases (85% or more). Italy, France and Spain were the countries that experienced major $NO_x$ emission reductions (down to 50%), a result that is in line with the strong lockdown restrictions implemented by their respective governments. On the contrary, Sweden, for example, showed reductions of only 15% (on $NO_x$) due to the implementation of national recommendations instead of a state-enforced lockdown. More details about the emission scaling procedure using the data and methodology from Guevara et al., 2020 can be found in Colette et al. (2020) where the resulting country and activity sector dependent reduction factors are provided for the EU28 countries plus Norway and Switzerland. Values of the emission reduction factors per country within the European regional modelling domain and per activity sector are provided in annex C. For the main contributing sector, road transport, the largest reductions in emissions are observed in countries where lockdown restrictions were more stringent (according to the Oxford Coronavirus Government Response Tracker stringency index Hale et al., (2020)), such as Italy (75%), Spain (80%) and France (76%).

All the models operated with the same setup as the CAMS regional operational production. The modelling domain covers Europe at $0.1° \times 0.1°$ resolution. The meteorological and chemical boundary conditions are obtained from the Integrated Forecasting System (IFS) of ECMWF, which is the same system that provides part of the dataset for the ML-based estimations (see sections 2 and 3). The baseline simulation was using the BAU anthropogenic emissions as described above and the lockdown scenario was using the lockdown-adjusted inventory, modulated by country and activity sectors. From the two sets of 11 model simulations, the median at each grid point is calculated from an ensemble simulation (as is routinely done for the operational CAMS predictions, Marecal et al., 2015). Differences between the BAU ensemble and the lockdown scenario ensemble are then used to calculate $NO_2$ reduction estimates.

## 4.2. Results

Figure 8 displays the relative change estimates for each European urban area defined in section 2.2. The estimates are calculated using the median of the full hourly distribution (Fig. 8a) and of the distribution at 'qa'-filtered S5P overpass times and dates only (Fig. 8b) for each urban area. As expected, urban areas in more stringent lockdown countries (i.e., Spain, Italy, France) show the largest reductions (e.g., down to 60% in Madrid, see Figure 9), whereas urban areas with less stringent

lockdown measures (i.e., Germany, Poland, Sweden) show smaller reductions (e.g., around 16% in Stockholm, see Figure 8). The time sampling difference (hourly versus S5P overpass) does not affect the model estimates much, only differences of a few per cent are seen for most of the European urban areas. On average, over the set of median estimates for each urban area, the difference is small with 30% for hourly estimates and 32% for S5P sampled estimates. This is expected as the emission reduction estimates used to generate the lockdown scenario ensemble are set constant over time (daily and hourly). This point is further expanded in the next section where model estimate results are compared to the other types of estimates.

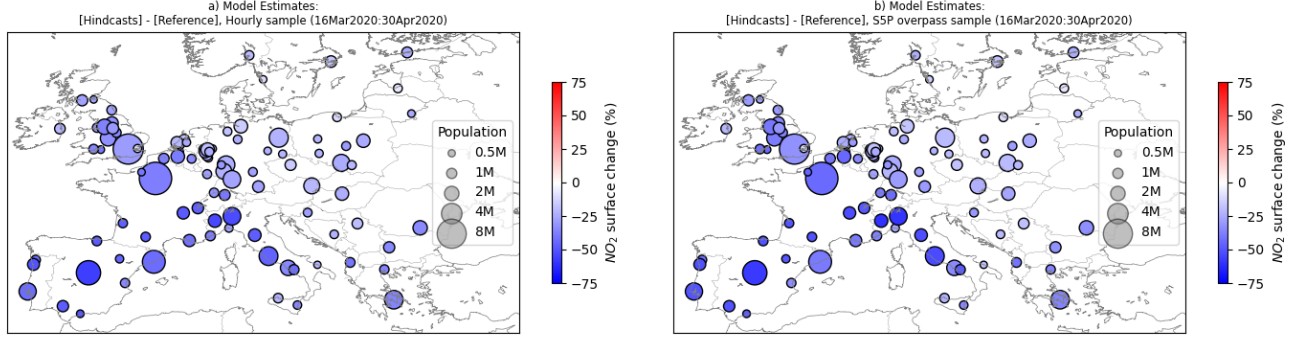

**Figure 8. Air quality modelling estimates of surface NO$_2$ changes (%, relative to the BAU predictions) during the lockdown period in urban areas with at least 0.5 million inhabitants. The left-hand side (a) shows the estimates using full hourly datasets and the right-hand side (b) shows the estimates using the S5P overpass time sampled dataset. The diameter of the circles is proportional to the population in each urban area.**

## 5. Comparison of the three different types of estimates

In table 4 and figure 9 we summarize the results of this study. Table 4 shows the average reduction of all the median estimates together with the inter-urban area variability over Europe. Figure 9 shows the distribution of the NO$_2$ changes estimated for the lockdown period per urban area. This figure provides estimates equivalent to box plots where the median and the inter-quartile range are displayed. For clarity, we chose to display only urban areas that are above 1 million inhabitants. The values of each estimate for all urban areas considered in this study are given in the table in annex B.

The three types of weather-normalised estimates agree on identifying stronger reductions where more severe lockdown measures were implemented. As shown in Section 2, satellite-based estimates show a relationship between NO$_2$ tropospheric column reductions and the extent and generalization of restrictive measures in each country. A similar relationship is observed for surface sites and model estimates (Sections 3 and 4). The largest NO$_2$ reduction estimates of around 50% to 60% for both surface and tropospheric columns are found in Spanish, Italian and French urban areas. In countries that implemented softer lockdown measures, urban areas show smaller reductions, e.g., Germany, Netherlands, Poland and

Sweden. Although significant discrepancies exist between the satellite-, surface- and model-based estimates in urban areas such as Naples (Italy), Sofia (Bulgaria), Katowice (Poland), the three methods provide an overall consistent picture. It is remarkable to note that this result contributes to establishing the usefulness of satellite-based estimates for urban air quality and not only for atmospheric pollution in general. Having a range of three different types of estimates helps to provide estimates of pollution changes across Europe with a certain level of certainty. When all the estimates agree, it is more likely that the values of reduction due to the lockdown implementations are reliable. Conversely, if the different types of estimates show discrepancies, less confidence should be given to the reduction estimates. In Fig. 8, Madrid, Turin and Milan, to mention a few urban areas, show consistency between the different types of estimates expressing more certainty in the results. In other locations such as Sofia, Athens and Budapest, strong discrepancies indicate that the estimates could be uncertain. Average scores in table 4 show that surface station observations provide stronger reduction estimates and that satellite-based estimates provide weaker reduction estimates. Model estimates are mostly in between and show much less spread within a given urban area (bars in Fig. 9) and less variation between urban areas (standard deviation in Table 4). The origin of such differences can vary and is detailed below.

Machine learning estimates that are observation-based (satellite and surface stations) are showing more spread compared to the model estimates. In Figure 9 the interquartile ranges for the observation-based ML estimates are much larger than for the model estimates. Such large ranges show that there is a strong spread in the ML-based estimates that is not seen in the model-based estimates. Model estimates are based on country-dependent emission reduction/scaling factors that are constant over time. The variability is induced by the changes in atmospheric conditions, but not by changes in the emissions. The estimates from the ML approach can represent the transition into the lockdown where emissions gradually decreased. This is contributing to the increased spread seen in the ML estimates. Scores from ML estimates (see table 1 and 3) also show significant RMSE that can add noise to the time series and add to the resulting spread of the distributions. A stronger spread in TROPOMI estimates is likely due to the small training set used. Disentangling the noise and the actual variability would need to be carefully done in future work.

All the different estimates presented in this study are consistent in their spatial scale using 0.1° × 0.1° TROPOMI averaged pixels that match the CAMS forecasts and surface stations within a 0.1° range from the city centre. Some of the smaller urban areas considered in this study likely have a footprint that is smaller than 0.1°, meaning that high pollution levels from the urban area are mixed with low pollution background levels. This could cause the pollution changes in the gridded estimates to be weaker than expected in certain urban areas (e.g., Katowice, Budapest, Glasgow, etc.). Also, even if the urban/suburban background stations are selected, the in-situ surface observations sample the pollution levels within a 0.1° × 0.1° pixel given their location. This sampling might not be exactly representative of the average pollution footprint within the same pixel. This average is the information given by the models or the satellites. These representativeness issues contribute to creating discrepancies between the type of estimates and hence generate uncertainty. The differences seen in Fig. 9 between surface station estimates and gridded estimates (models and satellites) point out such possible representativeness issues.

Representativeness is a difficult and important topic and deserves further research as it would require careful examination of
the stations' locations in specific urban areas and also using higher-resolution modelling than 10 km.

Satellite overpass times (13:30 local solar time) and the presence of clouds in the measurement pixel can potentially influence the reduction estimates from the TROPOMI data. We considered 1.5 months to compute the satellite reduction estimates. Overall, the sample size (S5P valid overpasses) in Fig. 9 ranges between 14 (Sevilla) and 37 (The Hague). In the same Figure 9, surface sites and model estimates are displayed for hourly and S5P sampled estimates. Smaller or larger samples
cannot really explain discrepancies between all the different estimates. Results, however, can be affected when the sample size becomes statistically very small and if shorter time periods (e.g., 1 or 2 weeks) are considered for satellite reduction estimates. Very small samples over the 6-week period were not considered in this study to avoid this effect. The sampling effect also shows greater changes in the surface station estimates than in the model estimates. As mentioned above and seen in Figure 9 the surface station estimates provide more variability that accounts for hourly variations. The model estimates have fixed
emission scaling factors for the entire lockdown period. The surface station estimates show more sensitivity to the time sampling than the model estimates. On average (see table 4), the S5P overpass sampling changes the estimates by around -6% for surface station estimates and only by -1.5% for model estimates. This suggests that the lockdown-induced reduction estimates depend upon the time of the day, i.e., those times when the road transport activity is peaking.

Finally, the reduction estimates for tropospheric $NO_2$ columns displayed in Figure 9 are generally not as strong as the
$NO_2$ surface estimates (observations and model). Some exceptions can be seen in certain Spanish (e.g., Barcelona, Madrid) and Italian (e.g., Milan, Turin) urban areas, where column estimates are close to the surface estimates, but overall column reductions are weaker. With all urban areas considered, the satellite estimates show around 23% reduction on average, which is 10% to 20% less than the model and surface station estimates (see table 4). This can be expected as $NO_2$ surface site measurements do not directly translate to the TROPOMI $NO_2$ tropospheric column, which is the integrated $NO_2$ content from
the surface to about the 200hPa altitude. Due to the short lifetime of $NO_2$ (around 12 hours), only small lockdown-induced changes to the free tropospheric $NO_2$ contents are expected. Changes are mainly expected near-surface and within the PBL. Therefore, the different nature of the vertical sampling is likely to contribute to the differences between the relative reduction estimates from tropospheric columns versus surface concentrations. Further work will be needed to link quantitatively the tropospheric column and surface-level variations, including sampling the model estimates using an observation operator
commonly used in data assimilation and inverse modelling systems. This important work will be carried out in a further study.

|  | Mean (%) | Standard Deviation (%) |
|---|---|---|
| Surface Stations [hourly] | -37 | 15 |
| Surface Stations [S5P sampling] | -43 | 19 |
| CAMS model ensemble [hourly] | -30 | 11 |
| CAMS model ensemble [S5P sampling] | -32 | 12 |

| | | |
|---|---|---|
| TROPOMI | -23 | 16 |

**Table 4. Scores over all European urban areas included in the study for the different NO₂ change estimates: based on surface observations, model estimates and TROPOMI observations. Mean and standard deviation are calculated for all urban area resulting estimates, i.e., the standard deviation is a metric of the inter-urban area spread.**

## 6. Conclusions


In this paper, we first show the importance of accounting for weather variability in satellite-based estimates of NO₂ changes due to the COVID-19 lockdown. While focusing on Europe and using the Sentinel-5p/TROPOMI instrument, we show that the satellite estimates based on direct comparisons between different time periods without accounting for weather variability can be flawed and should not be used for this kind of assessment. To account for weather variability in satellite

estimates, we use a recently developed methodology based on the gradient boosting machine learning technique. This methodology has proven to be efficient with surface sites to estimate lockdown induced changes over Spain (Petetin et al., 2020). We extended those surface estimates over Europe to compare with the satellite estimates. Finally, we included estimates of NO₂ changes using the 11-model CAMS regional ensemble, using emission reduction factors representative of the lockdown period. By providing and comparing the three different methodologies we provided a comprehensive and complimentary

assessment of NO₂ pollution level changes during the COVID-19 European lockdown. These assessments of pollution changes, when activity levels of key emitting sectors are significantly reduced (i.e., road transport and industry) in lockdown conditions, also provide crucial information to accurately quantify the benefits of the potential implementation of air quality policies for these emission sectors.

Main results show a consistent tendency of stronger reduction of NO₂ where more stringent lockdown measures were

implemented. On average, the three types of estimates show a reduction of 23%, 43% and 30% for satellite, surface stations and model estimates, respectively. Differences are explained by the different nature of the methods used, i.e., observation-based versus model-based, horizontal and vertical sampling, variability representation and time sampling. By providing an array of different methods we provide an indication of how reliable the pollution reduction estimates are for the various urban areas considered in this study. Accurately quantifying the pollution changes is also important for the impact of these pollution

reductions on the COVID-19 pandemic itself. Several studies have investigated the correlation between the high level of COVID-19 mortality and atmospheric pollution (e.g., Contincini et al. 2020, Ogen et al. 2020, Achebak et al., 2020). Feedbacks are then to be expected between the effects of short-term air pollution exposure on COVID-19 mortality and lockdown measures. Beyond the quantification of the impact of COVID-19-related restrictions on pollutant concentrations, the observation-based weather-normalisation methodology used in this study is of general interest for assessing the impact of any

type of emission changes (e.g., regulation and policy) on air quality (Grange et al., 2018, 2019) in the future.

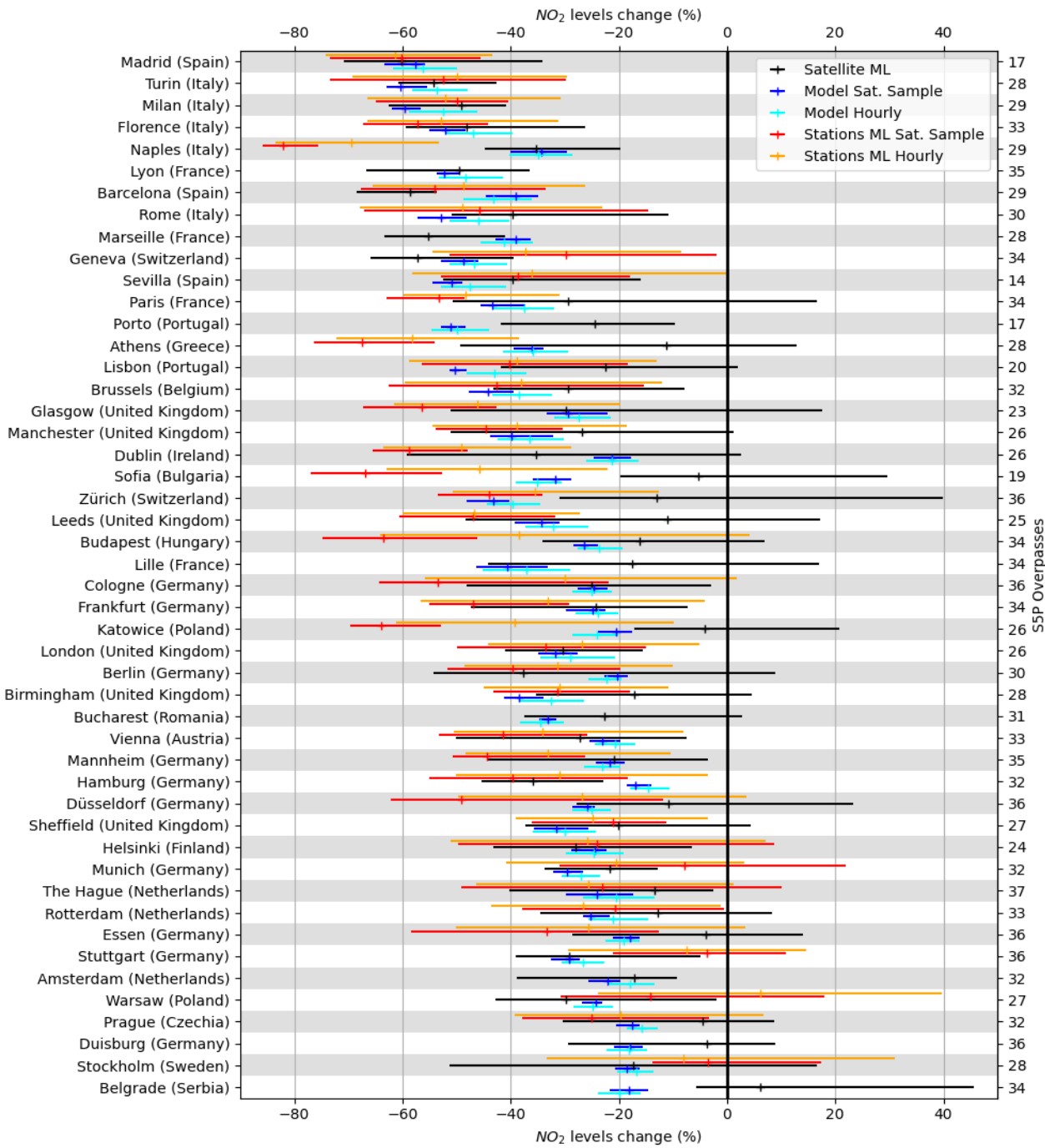

**Figure 9. Comparisons of the lockdown-induced NO₂ change estimates (%, relative to the BAU predictions) using different methodologies for European urban areas above 1 million inhabitants. Horizontal lines represent the**

**550** **interquartile ranges (over the temporal variability), and the ticks are the median values using the full distribution per urban area. For readability, urban areas are ranked using the average between all median estimates.**

**Acknowledgements**

The research leading to these results has received funding from the Copernicus Atmosphere Monitoring Service (CAMS), which is implemented by the European Centre for Medium-Range Weather Forecasts (ECMWF) on behalf of the European
**555** Commission. We acknowledge support from the Ministerio de Ciencia, Innovación y Universidades (MICINN) as part of the BROWNING project RTI2018-099894-B-I00 and NUTRIENT project CGL2017-88911-R, the AXA Research Fund and the 620 European Research Council (grant no. 773051, FRAGMENT). We also acknowledge PRACE and RES for awarding access to Marenostrum4 based in Spain at the Barcelona Supercomputing Center through the eFRAGMENT2 and AECT-2020-1- 0007 projects. This project has also received funding from the European Union's Horizon 2020 research and
**560** innovation programme under the Marie Sklodowska-Curie grant agreement H2020-MSCA-COFUND-2016-754433. Carlos Pérez GarcíaPando also acknowledges the support received through the Ramón y Cajal programme (grant RYC-2015-18690) of the MICINN. Modelling and satellite data were produced by the *Copernicus Atmosphere Monitoring Service.* We thank the 3 anonymous reviewers for their helpful comments that improved this paper.

**Annex A. Gradient Boosting Regressor Tuning**

**565** We have used TROPOMI data from 2019-01-01 to 2019-05-31 to train our machine learning simulator. We used the gradient boosting regressor function included in the scikit-learn python library. For validation purposes, the data set has been split between a training set (90% of the total dataset) and a test set (10% of the total dataset) using the train_test_split function. The hyperparameter tuning is then performed using the training set to generate the simulators and test set to find the best fit. Similarly, to Petetin et al. (2020) the learning rate was fixed to 0.05 and the number of features (max_features) is set to "sqrt".
**570** In addition, the tuning of the gradient boosting regressor was done for the following hyperparameters using the grid search method. The following hyperparameters were tuned: the subsample (subsample: from 0.3 to 1.0 by 0.1 with the best value of 0.9), the number of trees (n_estimators: from 50 to 1000 by 50 with the best value of 400) and the minimum sample in terminal leaves (min_samples_leaf: from 1 to 30 with the best value of 22). We use the default 5-fold cross-validation. We then test the final results on the test set in order to ensure not overfitting.

**575**

Links to the python libraries and functions:

Scikit-learn python

https://scikit-learn.org/stable/index.html

Gradient boosting function

**580** https://scikit-learn.org/stable/modules/generated/sklearn.ensemble.GradientBoostingRegressor.html

Grid search hyperparameter tuning

https://scikit-learn.org/stable/modules/generated/sklearn.model_selection.GridSearchCV.html

Random dataset splitting

https://scikit-learn.org/stable/modules/generated/sklearn.model_selection.train_test_split.html

**Annex B. Lockdown induced NO₂ changes estimates for each European urban area considered in this study**

| Urban Area | Country | TROPOMI Estimates (%) | N Revisits | Model Estimates Hourly (%) | Model Estimates S5P Sampled (%) | Surface Station Estimates Hourly (%) | Surface Station Estimates S5P Sampled (%) |
|---|---|---|---|---|---|---|---|
| Amsterdam | Netherlands | -17 | 32 | -18 | -22 | | |
| Antwerp | Belgium | -23 | 36 | -21 | -25 | -33 | -30 |
| Athens | Greece | -11 | 28 | -36 | -36 | -58 | -67 |
| Barcelona | Spain | -59 | 29 | -43 | -39 | -49 | -54 |
| Bari | Italy | -20 | 33 | -21 | -18 | -44 | -28 |
| Basel | Switzerland | -33 | 37 | -31 | -38 | -33 | -39 |
| Belgrade | Serbia | 6 | 34 | -20 | -18 | | |
| Berlin | Germany | -38 | 30 | -22 | -20 | -31 | -40 |
| Bilbao | Spain | -21 | 19 | -48 | -50 | -27 | -15 |
| Birmingham | UK | -17 | 28 | -33 | -38 | -31 | -31 |
| Bonn | Germany | -5 | 35 | -27 | -29 | -39 | -62 |
| Bordeaux | France | -22 | 28 | -47 | -50 | | |
| Bradford | UK | -24 | 26 | -31 | -34 | | |
| Braga | Portugal | -1 | 16 | -43 | -43 | | |
| Bremen | Germany | -37 | 34 | -18 | -20 | -37 | -49 |
| Brighton | UK | -22 | 31 | -21 | -24 | -23 | -27 |
| Bristol | UK | -19 | 30 | -40 | -44 | -38 | -39 |
| Brussels | Belgium | -29 | 32 | -38 | -44 | -38 | -43 |
| Bucharest | Romania | -23 | 31 | -34 | -33 | | |
| Budapest | Hungary | -16 | 34 | -24 | -26 | -38 | -64 |
| Bytom | Poland | -12 | 30 | -25 | -22 | | |
| Caerdydd | UK | -19 | 31 | -36 | -42 | -58 | -73 |
| Catania | Italy | -30 | 26 | -35 | -35 | | |
| Cologne | Germany | -25 | 36 | -25 | -25 | -30 | -53 |
| Dortmund | Germany | -11 | 36 | -24 | -24 | -29 | -48 |
| Dresden | Germany | -28 | 32 | -22 | -20 | -29 | -21 |
| Dublin | Ireland | -35 | 26 | -21 | -21 | -49 | -59 |

| | | | | | | |
|---|---|---|---|---|---|---|
| Duisburg | Germany | -4 | 36 | -18 | -18 | | |
| Düsseldorf | Germany | -11 | 36 | -25 | -26 | -27 | -49 |
| Edinburgh | UK | -16 | 23 | -28 | -28 | -39 | -34 |
| Essen | Germany | -3 | 36 | -19 | -18 | -26 | -33 |
| Florence | Italy | -48 | 33 | -47 | -52 | -53 | -57 |
| Frankfurt | Germany | -24 | 34 | -24 | -25 | -33 | -47 |
| Gdańsk | Poland | -17 | 30 | -11 | -10 | -23 | -43 |
| Geneva | Switzerland | -57 | 34 | -47 | -49 | -37 | -30 |
| Genoa | Italy | -36 | 30 | -27 | -27 | | |
| Glasgow | UK | -30 | 23 | -27 | -29 | -46 | -56 |
| Gliwice | Poland | -23 | 32 | -27 | -25 | | |
| Göteborg | Sweden | -5 | 32 | -10 | -14 | 8 | 19 |
| Hamburg | Germany | -36 | 32 | -15 | -17 | -31 | -40 |
| Hannover | Germany | -19 | 33 | -24 | -25 | -26 | -29 |
| Helsinki | Finland | -28 | 24 | -25 | -24 | -26 | -24 |
| Katowice | Poland | -4 | 26 | -24 | -20 | -39 | -64 |
| Kraków | Poland | -12 | 30 | -21 | -21 | -37 | -49 |
| Leeds | UK | -11 | 25 | -32 | -34 | -47 | -47 |
| Leipzig | Germany | -23 | 36 | -22 | -23 | | |
| Lille | France | -17 | 34 | -37 | -41 | | |
| Lisbon | Portugal | -22 | 20 | -43 | -50 | -39 | -40 |
| Liverpool | UK | -4 | 29 | -28 | | | |
| Liège | Belgium | 0 | 34 | -34 | -35 | -37 | -40 |
| Łódź | Poland | -12 | 30 | -29 | -29 | -24 | -38 |
| London | UK | -30 | 26 | -29 | -32 | -27 | -34 |
| Lyon | France | -49 | 35 | -48 | -52 | | |
| Madrid | Spain | -60 | 17 | -56 | -58 | -61 | -60 |
| Manchester | UK | -27 | 26 | -37 | -40 | -39 | -45 |
| Mannheim | Germany | -21 | 35 | -23 | -22 | -33 | -44 |
| Marseille | France | -55 | 28 | -41 | -39 | | |
| Milan | Italy | -49 | 29 | -52 | -59 | -52 | -50 |
| Munich | Germany | -22 | 32 | -27 | -30 | -21 | -8 |
| Málaga | Spain | 16 | 6 | -50 | -48 | -63 | -66 |
| Naples | Italy | -35 | 29 | -35 | -34 | -69 | -82 |
| Newcastle | UK | -30 | 22 | -27 | -30 | -42 | -54 |
| Nice | France | -34 | 24 | -38 | -37 | -59 | -61 |
| Nottingham | UK | -24 | 23 | -35 | -37 | -45 | -47 |
| Nuremberg | Germany | -7 | 31 | -27 | -28 | -39 | -46 |
| Oslo | Norway | -51 | 22 | -20 | -24 | | |

| | | | | | | |
|---|---|---|---|---|---|---|
| Palermo | Italy | -39 | 26 | -22 | -23 | | |
| Paris | France | -29 | 34 | -38 | -43 | -48 | -53 |
| Porto | Portugal | -24 | 17 | -50 | -51 | | |
| Poznań | Poland | -26 | 31 | -22 | -22 | -38 | -56 |
| Prague | Czechia | -4 | 32 | -16 | -18 | -20 | -25 |
| Riga | Latvia | 5 | 30 | -7 | -7 | -50 | -84 |
| Rome | Italy | -40 | 30 | -46 | -53 | -49 | -46 |
| Rotterdam | Netherlands | -13 | 33 | -21 | -25 | -27 | -21 |
| Rouen | France | -23 | 35 | -40 | -46 | | |
| Saarbrücken | Germany | -24 | 38 | -28 | -27 | -33 | -37 |
| Salerno | Italy | -32 | 26 | -43 | -48 | -62 | -57 |
| Sarajevo | Bosnia Herz. | -29 | 26 | -23 | -20 | | |
| Sevilla | Spain | -40 | 14 | -48 | -51 | -36 | -39 |
| Sheffield | UK | -20 | 27 | -30 | -32 | -25 | -21 |
| Sofia | Bulgaria | -5 | 19 | -35 | -32 | -46 | -67 |
| Southend | UK | -27 | 29 | -11 | -11 | -30 | -37 |
| Stockholm | Sweden | -17 | 28 | -17 | -18 | -8 | -3 |
| Stuttgart | Germany | -29 | 36 | -27 | -29 | -7 | -4 |
| The Hague | Netherlands | -13 | 37 | -21 | -24 | -26 | -23 |
| Thessaloníki | Greece | -32 | 27 | -36 | -36 | | |
| Tirana | Albania | -24 | 26 | -40 | -41 | | |
| Toulouse | France | -16 | 24 | -48 | -51 | | |
| Turin | Italy | -54 | 28 | -54 | -60 | -50 | -52 |
| Utrecht | Netherlands | -20 | 33 | -25 | -30 | -28 | -31 |
| Valencia | Spain | -34 | 22 | -35 | -33 | -63 | -71 |
| Vienna | Austria | -27 | 33 | -21 | -23 | -34 | -41 |
| Vilnius | Lithuania | 32 | 26 | -25 | -24 | -51 | -66 |
| Warsaw | Poland | -30 | 27 | -25 | -24 | 6 | -14 |
| Wiesbaden | Germany | -26 | 33 | -30 | -31 | -31 | -44 |
| Wrocław | Poland | -28 | 34 | -22 | -21 | -14 | -27 |
| Wuppertal | Germany | -13 | 36 | -25 | -25 | -27 | -39 |
| Zagreb | Croatia | -16 | 32 | -29 | -30 | -68 | -81 |
| Zaragoza | Spain | -8 | 27 | -45 | -49 | -47 | -49 |
| Zürich | Switzerland | -13 | 36 | -40 | -43 | -35 | -44 |

**Annex C. Reduction factors (%) by country and activity sector corresponding to the lockdown period over the modelled**
**European domain**

| Country | GNFR_B_Industry | GNFR_F_RoadTransport | GNFR_H_Aviation |
|---|---|---|---|
| Albania | -11.5 | -77 | |
| Austria | | -54 | -96 |
| Belarus | | -19 | |
| Belgium | -11.0 | -63 | -96 |
| Bosnia & Herz. | | -43 | |
| Bulgaria | -14.0 | -48 | -96 |
| Croatia | -21.5 | -65 | -93 |
| Czechia | -14.7 | -41 | -99 |
| Germany | -11.5 | -42 | -87 |
| Denmark | -17.3 | -40 | -97 |
| Estonia | -15.2 | -37 | -92 |
| Finland | -5.9 | -53 | -91 |
| France | -29.0 | -76 | -94 |
| Georgia | | -75 | |
| Great Britain | -21.0 | -67 | -88 |
| Greece | -14.9 | -66 | -91 |
| Hungary | -12.8 | -50 | -95 |
| Ireland | -12.6 | -64 | |
| Italy | -18.9 | -75 | -93 |
| Latvia | -12.7 | -35 | -99 |
| Lithuania | -13.4 | -47 | -100 |
| Luxembourg | -11.2 | -62 | -86 |
| Macedonia | -30.5 | -49 | -100 |
| Malta | | -48 | |
| Moldova | -21.5 | -57 | |
| Netherlands | -27.1 | -56 | -91 |
| Norway | -10.9 | -38 | -83 |
| Poland | -12.3 | -53 | |
| Portugal | -14.6 | -73 | |
| Romania | -10.2 | -62 | -100 |

| | | | |
|---|---|---|---|
| Russia | | -38 | |
| Serbia | | -57 | |
| Slovakia | -11.8 | -51 | -100 |
| Slovenia | -10.7 | -50 | -91 |
| Spain | -19.3 | -80 | -97 |
| Sweden | -12.4 | -31 | -95 |
| Switzerland | | -47 | -95 |
| Turkey | | -87 | |
| Ukraine | | -23 | |
| AVG (+other) | -15.5 | -54 | -94 |

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
