# Peer review of "Estimating lockdown induced European NO2 changes"

_Atmospheric Chemistry and Physics, 2020_

## Referee Comment (RC1) · Anonymous Referee #1 · 6 Nov 2020

Barré et al. (2020) describes estimates of the magnitude of NO2 reduction that can be attributed to COVID-19 lockdown measures in Europe in 2020. The paper is highly relevant given the continuation of COVID-19 and associated restrictions, interesting and well written. The argument for needing meteorological normalisation is, I think, sound, but the introduction would benefit from some more discussion why this is so, focussing on the role of different meteorological variables on NO2 concentration. The methodology employed to achieve the meteorological normalisation is sound for surface estimates. I think it would make much more sense to use surface mixing ratio estimates from TROPOMI rather than column values, given the fact that (a) the paper focuses on urban air quality where exposure by large populations is at the surface level and (b) surface temperature and wind are key predictor variables. The CAMS mod-

elling section is well presented, but so far lacks punch. It would substantially benefit the paper to discuss the emission reductions further as this would demonstrate not just that the NO2 did reduce, but why. Discussing the sectors responsible for emission reduction and therefore NO2 concentration reduction, and whether this is consistent across Europe, would provide the chemistry focus required for Atmospheric Chemistry and Physics.

Overall, I would recommend publication once these comments are addressed. Expanded comments on the above discussion are made below, followed by minor technical comments.

Expanded Comments:

1. The introduction needs to provide the reader with some more context on why studying NO2 during the lockdown is important. I think a paragraph could be included to this end, outlining the unique nature of this real world emissions-reduction experiment and its potential to help us understand potential broadscale impacts of future pollution reduction measures.

2. The introduction highlights that considering meteorology is important for NO2 prediction – I agree, but the introduction would benefit from a little more context on why this is the case. Just a couple of sentences are necessary on, e.g., boundary layer heights and NO:NO2 temperature dependent ratios that make this point clear.

3. Figure 2: Temperature is a really important factor for the NOx partitioning and warm temperature anomalies are highlighted in the text for early 2020. Therefore, I think Fig. 2 would benefit from showing the distribution of temperature as well as PBLH and wind speed.

4. Line 180-182: Please consider this sentence: "This illustrates that such "before-during" type of satellite comparisons is misleading and unfit for assessing the effects of COVID-19 lockdown because it is very sensitive to seasonal variations of weather
regimes and emissions."

What you have shown, to this point in the paper, is that different baselines provide different results for the 'lockdown NO2 change' and that the weather was different. You have not strictly proven, yet, the link between the two. At this point in the paper, you either need to prove the causal link with data, provide references for the statement 'because it is very sensitive. . ." or mute the sentence to something like "This illustrates that such 'before-during' satellite comparisons clearly provide very different results as to the effect of lockdown on European NO2. This led us to investigate weather considering meteorology may provide a more consistent picture".

5. Why did you use TROPOMI column NO2 rather than surface mixing ratios? The mixing ratio can be determined from the NO2 tropospheric vertical profile. This would seem to me more relevant for urban air quality than tropospheric column values, and would provide consistency with surface observations. In addition, this would make your surface meteorological predictor values much more relevant, at the moment (effectively) surface temperature and wind are being used to predict the whole column.

6. Why were NO2 modelled concentrations in the predictor variables? I think to demonstrate the importance of meteorological normalisation, you should show that the GBM gives good prediction independent of NO2 concentration estimates.

7. Table 3: Please clarify whether the average changes in this table are means or medians, ideally consistent across all comparisons.

8. Line 283-284: "Using the last three years is long enough to capture weather variability at each site, but not too long with regards to long-term reduction of NO2 happening as a result of policy measures across Europe" – perhaps the authors could clarify this rather vague statement by indicating expected (or citing known) NO2 trends across Europe as a result of policy measures. In the previous paragraph, the authors note that their method underestimates NO2 in the pre-lockdown period by 8 %, could this be partly due to decreasing NO2 trends driven by policy or emissions?

9. I think the strength of the paper would be improved by discussing in more detail what led to the NO2 decrease – the modelling section seems to offer this opportunity. Was it reduction in industry, aviation, road transport, all of the above equally or something else that was primarily responsible for the NO2 change, and was this consistent across Europe?

10. It would be interesting to discuss how this kind of weather-normalisation 'business as usual' prediction could be implemented for air quality forecasting, in addition to event/emission change analysis.

Technical comments:

1. Be consistent with subscripting of "x" in NOx

2. Line 72: please revise the first sentence of the paragraph, 'very changing' is poor grammar – perhaps 'highly variable' would be better

3. I'm not sure Table 1 is necessary, it is so small and the information is clearly stated in the text anyway.

4. Line 123: strange font difference in 0.1x0.1o

5. Figure 3 (and subsequent similar figures): Given that you consider urban areas down to 0.5 million inhabitants, I recommend adding some more circles to your population circle-size legend (maybe 0.5 m, 1 m, 2 m, 4 m, 8 m)

6. Figure 3: (and subsequent similar figures) Subscripts please on the NO2 in the colour bar label

7. Figure 3 (and subsequent similar figures): please just clarify, the % change is relative to each baseline scenario? – I suggest including this clarification in the figure caption.

8. Table 2/line 245: are the outliers included in the statistics presented in table 2? If they're included, might they explain the significant RMSE?

9. Lines 263-265: Perhaps my personal choice, but I would write "X % reduction" not the double negative "-X% reduction". This would also be consistent with the way it is written in the paragraph starting Line 347.

10. Line 307: should be "...measurements do not directly translate to..."

11. Line 323: model rather than models

12. Line 338: I'm curious if there is a metric which could help determine the stringency of lockdown measures in different countries? At the moment, knowledge of the scale of lockdowns and COVID-19 consequences are fresh in our minds, but people may not have a feel for that reading this in the future. I think some discussion of what constitutes a 'more stringent' vs 'less stringent' lockdown is warranted.

---

## Referee Comment (RC2) · Anonymous Referee #2 · 6 Nov 2020

This paper addresses the impact of the European 2020 Covid lockdown on NO2 levels using satellite data, surface NO2 data and model simulations. Because of the short TROPOMI satellite record the impact of meteorology is derived using a machine-learning algorithm, which is also applied to the surface data.

The large impact of meteorological differences between 2019 and 2020 is noted and this serves as a caveat to some previous simple presentations of the data during/after lockdown. The impact of lockdown is quantified for all large European cities by the 3 methods. These are important and useful results to publish, in a timely manner given the interest in the impact on the effects of lockdown.

I only have minor comments and I think that the paper is publishable. My main comment is that the reader does not get a feel for the ML methodology and how well it works Printer-friendly version

pictorially. Text refers to large outliers but it is important to show this to the reader (see my comment on Figure 5).

The paper is readable as is, but there is a very large number of minor grammatical errors which will need addressing. Maybe the ACP office will do that. I don't have time to go through them all, but I would point out that the typos start in affiliation 1 for the lead author (Forecasts not Forecast and Shinfield not Sinfield!). Not a good start. In fact the errors start in the paper title (which would need a hyphen: lockdown-induced).

Other Specific Comments

Section 2.1 line 115. Give the local time of the TROPOMI observations in this section.

Line 197. 'not expected'. Make it clear that this is not expected based on emissions. One could expect this if one understood the impact of meteorology.

Line 225. 'Contrary to' change to 'In contrast to...'

Line 231. Table 1. Spell out the acronyms in the table headings.

Line 239-240. Explain what is meant by 'overfitting', what the implications would be and how you know it is not occurring.

Figure 5 shows median values and not the mean. How different would Figures 3 and 4 be if the median was used? This needs some more explanation and somehow the same methodology should be included in one of the cases. You could make Figure 5 into 4 panels and show both methods. It is important to show the limitations of the ML method and provide the equivalent results to the other methods.

Line 266. 'would be expected'. How large is the interannual variability on NO2 emissions?

Line 288. 'Contrary to' -> 'in contrast'.

Line 294. Same comment as above on overfitting.
Line 312. What does 'marginal' mean here? Small? Better to say what the lifetime of NO2 is and say that the impact is likely small.

Figure 6 caption. Say that these data are weather-normalized.

---

## Referee Comment (RC3) · Anonymous Referee #3 · 12 Nov 2020

**Review of Barré et al., Estimating lockdown induced European NO2 changes, submitted to ACP, 2020**

The authors quantify the reduction in NO2 levels over Europe that resulted from the decline in emitting activity during the Spring 2020 lockdowns, themselves resulting from government responses to the COVID19 pandemic. They do this by using satellite NO2 column data, surface measurements and model simulations, while also demonstrating the importance of accounting for year-to-year variability in weather conditions that would otherwise influence the NO2 signal on top of any emission changes. They conclude with a brief synthesis and comparison of the different methods, showing the estimated NO2 reductions for large European urban areas.

Overall, this is an interesting application and demonstration of state-of-the-art measurement, analysis and modelling tools to a timely topic. My questions, comments and concerns about the science are minor and outlined below. However, my main issue is more around presentation and structure. To me, the manuscript currently reads like a series of disconnected stories that are only weakly united at the end, with a rather thin discussion and summary. I expand on this comment and make some suggestions below, but I think addressing it would be a sizeable task (hence suggesting "major revisions"). I would urge the authors to consider this point since I think it would ultimately leave them with a much more readable (and citable!) piece of research.

**Major comments – Structure, presentation and focus**
A key selling point of this research is the multiple approaches that the authors have applied, yet this is not really front and centre to the reader, except in the Abstract. I would suggest reflecting this contribution in the title (e.g., "Lockdown-induced $NO_2$ reductions in Europe estimated from satellites, surface stations and air quality models"??) as well as in the first paragraph of the introduction. Currently, the introduction is rather focussed on reporting individual lockdown studies (which can probably be synthesised more) and discussing actual and potential misapplications of TropOMI data. There is not much information or discussion on what can be gleaned from surface observations and models, let alone why an approach with all three might be novel and more robust.

I would suggest that the authors then consider the presentation of the methods and results. One way would be to describe the measurement and model details and analysis approaches in one section, followed by a results section that begins with the current Figure 8 (which is the main take home message). Subsequent sections could then explore the differences between the approaches (e.g., combining some of the other maps?) as well as highlighting what are more well-known or secondary aspects, such as the need to consider meteorological normalisation. A final discussion section could consider the uncertainties in each approach in more detail.

Even if the above suggestion is not followed, the interpretation and discussion around the current Figure 8 certainly needs more attention and discussion. The submitted manuscript is rather scant on detail in comparing the outcome of the different approaches, how independent they are (e.g., are the model or surface measurements used in the satellite retrieval method or validation?), or how they may be used to provide some validation of each other or increase the overall confidence (e.g., as per IPCC type language like "very likely" etc. when there are several lines of evidence).

Finally, related to the presentation, I would encourage the authors to revisit the readability/flow and grammar of the manuscript. For the former, I often found that

paragraphs did not nicely follow on from one another, reading instead like disparate bullet points. Additionally, many longer paragraphs could be broken down into more readable chunks. Regarding grammar, to my mind there are several examples of curious word choice and word order. I accept that this may just be my preference coming through, but I would encourage the authors to proofread any resubmission.

**Specific comments**

Introduction: Somewhere, I would find space to acknowledge earlier work on weather normalization of AQ observations. One suggestion (but not limited to this!) is David Carslaw, whose blogs on the impact of lockdowns on NO2 refer to his published work (e.g., see: https://ee.ricardo.com/news/blog-update-on-covid-19-and-changes-in-air-pollution)

L108: Here or elsewhere (methods or results?) it would be good to be explicit about the otherwise implicit assumptions about BAU – i.e., that you're assuming emitting activity would be similar to previous years (for the weather normalized techniques), or as per the projected 2020 emissions data (for the simulations…although are these indeed be the same as previous years?).

Table 1: There is really no point in this Table, whose information could just be included in the text.

L143: How was the PBL height calculated and/or where did it come from?

L168: What are the criteria for "urban areas"? I am curious because it seems that the definition must include some of the surrounding metro areas (e.g., Southend, Essex, UK "proper" has a population < 200k), yet some major areas are excluded (e.g., the South Hampshire metro area in the UK has a population >1M).

Section 2.3: A figure showing the performance of the ML technique would be helpful. E.g., time series for a particular location, showing its performance for the training and test data sets?

L264: I'm not sure what "perform better" means here.

L284: Provide citation for "policy measures across Europe"

L300: I'm confused by this sentence – is it related to comparing the surface observations against the satellite data? Please clarify.

Section 4: However this section gets worked into a revised manuscript, more information is needed, even if it just points to other studies. I would encourage separate sections on the modeling set up and the emissions, as well as how the activity data (etc) were used. Also, how does the model output compare to TropOMI?

L321: Do the 11 models need to be named? Perhaps just point to the citation?

Section 5: As noted above, this section needs more discussion on Figure 8 and the difference between the results. Some additional specific comments follow:

L384: Explain/justify why it is "crucial…for air quality policy".

L389: What is meant by "relevance" here? I would argue is more of convenience, since the plot will be missing out a large majority of Europe's total population!

L412: Explain/describe "background footprint" and clarify the "representativeness" issues for more general readers.

Figure 8: This is a great figure, but it is rather busy with the lines which prevents any clear message emerging from a glance. Hard to know what to suggest (put hourly station and model data in an appendix figure, so it's comparing like with like?), but I would at least encourage the authors to make the zero line more obvious.

Figure 8: A separate issue to the above, the spread (IQR etc) needs a clearer definition. Is it a spatial and temporal spread?

**Technical corrections (a full proofread is recommended)**

L45: "…part of the nitrogen oxides…" – nitrogen oxides include a lot more than NO and NO2 (e.g., N2O, N2O5 etc). To me this is also an example of curious wording. Suggest "Nitrogen dioxide (NO2; together with NO, a constituent of NOx) is a very well-established…"

L72: "The storm Ciara..." -> "Storm Ciara…" (and in other cases too). I'm no expert but seems like the preferred orthography is to capitalize the "S" in Storm when referring to a named one.

L140: "A number of **named** extratropical cyclones (**Storms** Ciara, …)"

L269: This is an example long paragraph that could be broken into shorter ones.

L326: Spell out TNO

L399: This sentence doesn't make sense

L429: This is not a stand-alone sentence (belongs as a clause of previous sentence).

---

## Author Comment (AC1) · 19 Jan 2021

*We would like to thank the reviewer for their comments that helped to improve the paper's quality. Please read our answers in italic fonts below.*

Barré et al. (2020) describes estimates of the magnitude of NO2 reduction that can be attributed to COVID-19 lockdown measures in Europe in 2020. The paper is highly relevant given the continuation of COVID-19 and associated restrictions, interesting and well written. The argument for needing meteorological normalisation is, I think, sound, but the introduction would benefit from some more discussion why this is so, focussing on the role of different meteorological variables on NO2 concentration. The methodology employed to achieve the meteorological normalisation is sound for surface estimates. I think it would make much more sense to use surface mixing ratio estimates from TROPOMI rather than column values, given the fact that (a) the paper focuses on urban air quality where exposure by large populations is at the surface level and (b) surface temperature and wind are key predictor variables. The CAMS modelling section is well presented, but so far lacks punch. It would substantially benefit the paper to discuss the emission reductions further as this would demonstrate not just that the NO2 did reduce, but why. Discussing the sectors responsible for emission reduction and therefore NO2 concentration reduction, and whether this is consistent across Europe, would provide the chemistry focus required for Atmospheric Chemistry and Physics.
Overall, I would recommend publication once these comments are addressed. Expanded comments on the above discussion are made below, followed by minor technical comments.

*Unfortunately, the TROPOMI instrument as a nadir viewing geometry does not provide surface mixing ratio retrievals but only tropospheric columns retrievals. Due to the nature of the measurements (reflected radiation from earth surface in UV-Visible part of the spectrum in the case of TROPOMI) is it very well established in the scientific community that such type of remote sensing measurements cannot provide retrieved surface concentration solely but only vertically integrated retrieved content with provided weighting function called the averaging kernel.*

*Regarding the emission suggestion, we state in the introduction: "These lockdowns drastically reduced traffic and also activity levels in most industries (Guevara et al., 2020; Le Quéré et al., 2020). These sectors represent a large share of $NO_x$ emissions (51% according to EEA 2020a)." The references mentioned in the text already discuss extensively the sectors for emission reductions and also show the large share of $NO_x$ emission in the impacted sectors. This is recalled in the modelling section where the same references are mentioned and also adding Colette et al. (2020) that refers on how this was implemented in the modelling procedure. We do not think then adding extensive discussion from already published results in the literature will be an added value to the article, but we now add more description on the emission reduction estimates, recalling the information given by Guevara et al., 2020 and Colette et al., 2020 in the modelling section and adding a table showing the emission inventory reduction factors used per country and per sector.*

Expanded Comments:

1. The introduction needs to provide the reader with some more context on why studying NO2 during the lockdown is important. I think a paragraph could be included to this end, outlining the unique nature of this real world emissions-reduction experiment and its potential to help us understand potential broadscale impacts of future pollution reduction measures.

*We have complemented the introduction accordingly.*

2. The introduction highlights that considering meteorology is important for NO2 prediction – I agree, but the introduction would benefit from a little more context on why this is the case. Just a couple of sentences are necessary on, e.g., boundary layer heights and NO:NO2 temperature dependent ratios that make this point clear.

*We have detailed the introduction accordingly.*

3. Figure 2: Temperature is a really important factor for the NOx partitioning and warm temperature anomalies are highlighted in the text for early 2020. Therefore, I think Fig. 2 would benefit from showing the distribution of temperature as well as PBLH and wind speed.

*We have added the temperature distributions in the figure.*

4. Line 180-182: Please consider this sentence: "This illustrates that such "beforeduring" type of satellite comparisons is misleading and unfit for assessing the effects of COVID-19 lockdown because it is very sensitive to seasonal variations of weather regimes and emissions." What you have shown, to this point in the paper, is that different baselines provide different results for the 'lockdown NO2 change' and that the weather was different. You have not strictly proven, yet, the link between the two. At this point in the paper, you either need to prove the causal link with data, provide references for the statement 'because it is very sensitive. . ." or mute the sentence to something like "This illustrates that such 'before-during' satellite comparisons clearly provide very different results as to the effect of lockdown on European NO2. This led us to investigate weather considering meteorology may provide a more consistent picture".

*We have clarified the sentence.*

5. Why did you use TROPOMI column NO2 rather than surface mixing ratios? The mixing ratio can be determined from the NO2 tropospheric vertical profile. This would seem to me more relevant for urban air quality than tropospheric column values, and would provide consistency with surface observations. In addition, this would make your surface meteorological predictor values much more relevant, at the moment (effectively) surface temperature and wind are being used to predict the whole column.

*Please see the response above. In addition, the surface mixing ratio cannot be straightforwardly determined from TROPOMI measurement. We are not sure what the reviewer means by "The mixing ratio can be determined from the NO2 tropospheric vertical profile."? Then which NO2 tropospheric vertical profile? From measurements (in situ I presume), from models? Column based quantities are retrieved from the TROPOMI measurements not profiles.*

6. Why were NO2 modelled concentrations in the predictor variables? I think to demonstrate the importance of meteorological normalisation, you should show that the GBM gives good prediction independent of NO2 concentration estimates.

*Including the business-as-usual $NO_2$ surface concentrations from the CAMS regional forecasts will help the GB model to simulate more accurately $NO_2$ columns where the meteorological information solely is not enough as the training set is very small. Using the business-as-usual CAMS $NO_2$ forecasts or not in the predictors do not interfere with the validity of the meteorological normalisation method. We used business as usual predictors only and whether they are meteorological information, chemical forecast information or else do not change the methodology to predict $NO_2$ columns to compare with the factual or real $NO_2$ columns. We now clarify that the $NO_2$ forecast information used is business as usual information.*

7. Table 3: Please clarify whether the average changes in this table are means or medians, ideally consistent across all comparisons.

*We have clarified the caption.*

8. Line 283-284: "Using the last three years is long enough to capture weather variability at each site, but not too long with regards to long-term reduction of NO2 happening as a result of policy measures across Europe" – perhaps the authors could clarify this rather vague statement by indicating expected (or citing known) NO2 trends across Europe as a result of policy measures. In the previous paragraph, the authors note that their method underestimates NO2 in the pre-lockdown period by 8 %, could this be partly due to decreasing NO2 trends driven by policy or emissions?

*Since GBM models are trained based on past data, strong trends between the training period (2017-2019) and the period of interest (2020) may indeed affect the reliability of the ML-based predictions. In order to limit this issue, we introduced the date index feature. If no substantial trend exists since 2017, the GBM models will not use this feature and will benefit from the entire (2017-2019) training dataset for learning the influence of the meteorology on NO2 surface mixing ratios. Conversely, if a strong trend is present, the GBM models will give more importance to this date index feature during their training, which in practise will allow predicting NO2 mixing ratios (in 2020) in the range of values observed during the last part of the training dataset. This is discussed in more detail in Petetin et al. (2020). We clarified the statement and also added a reference for trend in the text.*

9. I think the strength of the paper would be improved by discussing in more detail what led to the NO2 decrease – the modelling section seems to offer this opportunity. Was it reduction in industry, aviation, road transport, all of the above equally or something else that was primarily responsible for the NO2 change, and was this consistent across Europe?

*Please see the response to the main comment above. The modelling section has been complemented accordingly.*

10. It would be interesting to discuss how this kind of weather-normalisation 'business as usual' prediction could be implemented for air quality forecasting, in addition to event/emission change analysis.

*To reinforce the usefulness of the method on observation-based emission change analysis, we added the following sentence in the conclusion: "Beyond the quantification of the impact of COVID-19-related restrictions on pollutant concentrations, the weather-normalization methodology used in this study is of general interest for assessing the impact of any type of emission changes (e.g. regulation) on air quality (Grange et al., 2018, 2019)."*

*It is important to note that, state of the art air quality forecasting is using models that do include the meteorological variability intrinsically in their predictions as such models represent and predict explicitly the evolution of the pollutants in the atmosphere. So, no need of weather normalisation here. CAMS operationally run such forecast models daily at 10km resolution over Europe. In addition, data assimilation is performed within those models to include the information from observations in an optimal way. With the current methodology presented here BAU parameters coming from air quality and numerical weather prediction models are needed. So, using such weather normalized technique to forecast air quality will be redundant. One of the advantages and currently looked at application of machine learning techniques on air quality forecasting focuses more on designing downscaling methods to predict phenomena beyond the model's resolution, but this is rather out of the scope of the paper.*

Technical comments:

1. Be consistent with subscripting of "x" in NOx

*Fixed*

2. Line 72: please revise the first sentence of the paragraph, 'very changing' is poor grammar – perhaps 'highly variable' would be better

*We have corrected the sentence*

3. I'm not sure Table 1 is necessary, it is so small and the information is clearly stated in the text anyway.

*We have removed table 1*

4. Line 123: strange font difference in 0.1x0.1o

*The fonts have been changed*

5. Figure 3 (and subsequent similar figures): Given that you consider urban areas down to 0.5 million inhabitants, I recommend adding some more circles to your population circle-size legend (maybe 0.5 m, 1 m, 2 m, 4 m, 8 m)

*We have adjusted the plots*

6. Figure 3: (and subsequent similar figures) Subscripts please on the $NO_2$ in the colour bar label

*We have adjusted the plots*

7. Figure 3 (and subsequent similar figures): please just clarify, the%change is relative to each baseline scenario? – I suggest including this clarification in the figure caption.

*We have clarified the captions.*

8. Table 2/line 245: are the outliers included in the statistics presented in table 2? If they're included, might they explain the significant RMSE?

*Yes, a strong RMSE in the performance results will most likely generate outlier in the $NO_2$ changes estimates. We clarify the text accordingly.*

9. Lines 263-265: Perhaps my personal choice, but I would write "X % reduction" not the double negative "-X% reduction". This would also be consistent with the way it is written in the paragraph starting Line 347.

*Corrected*

10. Line 307: should be ". . .measurements do not directly translate to. . ."

*Corrected*

11. Line 323: model rather than models

*Corrected*

12. Line 338: I'm curious if there is a metric which could help determine the stringency of lockdown measures in different countries? At the moment, knowledge of the scale of lockdowns and COVID-19 consequences are fresh in our minds, but people may not have a feel for that reading this in the future. I think some discussion of what constitutes a 'more stringent' vs 'less stringent' lockdown is warranted.

*There is such a metric developed by Oxford university. It accounts for various parameters to define a stringency index. We now mention this index with the reference in the text.*

---

## Author Comment (AC3) · 19 Jan 2021

**Review of Barré et al., Estimating lockdown induced European NO2 changes, submitted to ACP, 2020**

Anonymous Referee #3

*We would like to thank the reviewer for their comments that helped to improve the paper's quality. Please read our answers in italic fonts below.*

The authors quantify the reduction in NO2 levels over Europe that resulted from the decline in emitting activity during the Spring 2020 lockdowns, themselves resulting from government responses to the COVID19 pandemic. They do this by using satellite NO2 column data, surface measurements and model simulations, while also demonstrating the importance of accounting for year-to-year variability in weather conditions that would otherwise influence the NO2 signal on top of any emission changes. They conclude with a brief synthesis and comparison of the different methods, showing the estimated NO2 reductions for large European urban areas.
Overall, this is an interesting application and demonstration of state-of-the-art measurement, analysis and modelling tools to a timely topic. My questions, comments and concerns about the science are minor and outlined below. However, my main issue is more around presentation and structure. To me, the manuscript currently reads like a series of disconnected stories that are only weakly united at the end, with a rather thin discussion and summary. I expand on this comment and make some suggestions below, but I think addressing it would be a sizeable task (hence suggesting "major revisions"). I would urge the authors to consider this point since I think it would ultimately leave them with a much more readable (and citable!) piece of research.

**Major comments – Structure, presentation and focus**

A key selling point of this research is the multiple approaches that the authors have applied, yet this is not really front and centre to the reader, except in the Abstract. I would suggest reflecting this contribution in the title (e.g., "Lockdown-induced NO2 reductions in Europe estimated from satellites, surface stations and air quality models"??) as well as in the first paragraph of the introduction.

*We have changed the title.*

Currently, the introduction is rather focussed on reporting individual lockdown studies (which can probably be synthesised more) and discussing actual and potential misapplications of TropOMI data. There is not much information or discussion on what can be gleaned from surface observations and models, let alone why an approach with all three might be novel and more robust.

*The information about surface observations and models were already there but it was rather disorganised. We have now re-arranged and re-written the introduction. Synthesised some paragraphs (i.e., satellite) and adding more information on others (e.g., surface observations). It is now more in accordance with the rest of the paper.*

I would suggest that the authors then consider the presentation of the methods and results. One way would be to describe the measurement and model details and analysis approaches in one section, followed by a results section that begins with the current Figure 8 (which is the main take home message). Subsequent sections could then explore the differences between the approaches (e.g., combining some of the other maps?) as well as highlighting what are more well-known or secondary aspects, such as the need to consider meteorological normalisation. A final discussion section could consider the uncertainties in each approach in more detail.

*We understand the reviewers' point of view, but we cannot operate such changes at this stage of the paper's submission. This will completely break the flow of explanations and will require re-writing almost entirely the paper on top of the already major changes requested in this review. We however have reorganised the three section on satellite observation, surface observations and air quality models with methods and results subsections. We now split the former "Summary and discussion" section into two: one section about the comparisons of the three approaches (former figure 8 now figure 9 discussion) and one general and final conclusion.*

Even if the above suggestion is not followed, the interpretation and discussion around the current Figure 8 certainly needs more attention and discussion. The submitted manuscript is rather scant on detail in comparing the outcome of the different approaches, how independent they are (e.g., are the model or surface measurements used in the satellite retrieval method or validation?), or how they may be used to provide some validation of each other or increase the overall confidence (e.g., as per IPCC type language like "very likely" etc. when there are several lines of evidence).

*We have now extended the discussions for each of the individual results and the comparisons around the final figure (now figure 9). An entire section is now devoted to the comparisons of the different methods and provide extended explanation of why they could differ or not. The section expands over two pages.*
*Note that the NO2 forecasts used in the training set and predictors is business as usual information as no assimilation performed therefore making it independent form the model estimates provided in section 4. We clarified the text accordingly.*

Finally, related to the presentation, I would encourage the authors to revisit the readability/flow and grammar of the manuscript. For the former, I often found that paragraphs did not nicely follow on from one another, reading instead like disparate bullet points. Additionally, many longer paragraphs could be broken down into more readable chunks. Regarding grammar, to my mind there are several examples of curious word choice and word order. I accept that this may just be my preference coming through, but I would encourage the authors to proofread any resubmission.

*The flow and grammar have been revised.*

**Specific comments**

Introduction: Somewhere, I would find space to acknowledge earlier work on weather normalization of AQ observations. One suggestion (but not limited to this!) is David Carslaw, whose blogs on the impact of lockdowns on NO2 refer to his published work (e.g., see: https://ee.ricardo.com/news/blog-update-on-covid-19-and-changes-in-air-pollution)

*The references were added in the introduction with few additional sentences. Note that the (published) work of David Carslaw is acknowledged through the references to Grange et al. (2018, 2019) (Carslaw being the second author of these studies).*

L108: Here or elsewhere (methods or results?) it would be good to be explicit about the otherwise implicit assumptions about BAU – i.e., that you're assuming emitting activity would be similar to previous years (for the weather normalized techniques), or as per the projected 2020 emissions data (for the simulations…although are these indeed be the same as previous years?).

*This point was answered in the other referee comments. We now refer to the EEA 2020a report in the satellite and surface observation sections to acknowledge the potential contribution of the NOx emission trend in the BAU predictions.*

*Reference:*
*EEA: European Union emission inventory report 1990-2018 under the UNECE Convention on Long-range Transboundary Air Pollution (LRTAP), EEA Report No 05/2020, 2020a.*

Table 1: There is really no point in this Table, whose information could just be included in the text.

*We have removed the table*

L143: How was the PBL height calculated and/or where did it come from?

*The PBL height comes from the IFS calculations. We clarified the text accordingly and provide the reference to the IFS documentation for calculation details.*

L168: What are the criteria for "urban areas"? I am curious because it seems that the definition must include some of the surrounding metro areas (e.g., Southend, Essex, UK "proper" has a population < 200k), yet some major areas are excluded (e.g., the South Hampshire metro area in the UK has a population >1M).

*We have used the free data base coming from (https://simplemaps.com/data/world-cities). Following the website description and Q&A:*

*" What counts as a city/town?*

*Any populated place in the world as determined by U.S. government agencies. Neighborhoods within listed cities are not included.*

*Where does your data come from?*

*Cities for all non-U.S. countries comes from the National Geospatial-Intelligence Agency. US cities data comes from the U.S. Census Bureau and the U.S. Geological Survey. The basic database and population data comes from Natural Earth Data. Population density data comes from the Center for International Earth Science Information Network at Columbia University in partnership with NASA's Socioeconomic Data and Applications Center. We've made a concerted effort to source our data from public domain and permissively-licensed sources that will not restrict the rights of our customers."*

*So, it is possible that cities boundary definitions are not reflecting the overall population in the urban area. We choose to use the term urban area as for many cases a pixel of 10 km x 10 km is not representative of a given city but also of its surroundings. We now provide the reference site in the text for the database that has been used and clarified the sentence.*

Section 2.3: A figure showing the performance of the ML technique would be helpful. E.g., time series for a particular location, showing its performance for the training and test data sets?

*We have added the suggested additional figure providing times series for 2019 and 2020 for Madrid. We also have amended the text to include the figure description and discussion.*

L264: I'm not sure what "perform better" means here.

*The sentence has been clarified.*

L284: Provide citation for "policy measures across Europe"

*The reference is now included in the text.*

L300: I'm confused by this sentence – is it related to comparing the surface observations against the satellite data? Please clarify.

*Yes, it is. We have clarified the text.*

Section 4: However this section gets worked into a revised manuscript, more information is needed, even if it just points to other studies. I would encourage separate sections on the modeling set up and the emissions, as well as how the activity data (etc) were used. Also, how does the model output compare to TropOMI?

*From the other reviewer comments, we have clarified the emission scaling procedure with the necessary references. We also provided the country dependent reductions factors in the annex section.*

*The study focuses on the assessment of the relative reductions seen by various methods but not an evaluation and validation of the CAMS regional models using TROPOMI. Such activity of validation using surface data and satellite data is routinely performed quarterly on the operational production and delivered publicly. We have added clarifications and the link to the validation reports in the text.*
*A fully consistent comparison between model and TROPOMI estimates would require a satellite observation operator. Unfortunately, such operator is not officially available yet for the regional models. This point is discussed in the summary and discussion section of the first submitted version of the paper. This is now in the last paragraph of section 5.*

L321: Do the 11 models need to be named? Perhaps just point to the citation?

*The model names have been removed.*

Section 5: As noted above, this section needs more discussion on Figure 8 and the difference between the results. Some additional specific comments follow:

L384: Explain/justify why it is "crucial…for air quality policy".

*The text has been clarified.*

L389: What is meant by "relevance" here? I would argue is more of convenience, since the plot will be missing out a large majority of Europe's total population!

*We have removed the word relevance. It is however in the most densely populated areas that the pollutions changes are expected to be seen at a 10 km scale. Business as usual pollution levels have the tendency to be higher in large urban areas than small urban areas.*

L412: Explain/describe "background footprint" and clarify the "representativeness" issues for more general readers.

*We have clarified the text accordingly.*

Figure 8: This is a great figure, but it is rather busy with the lines which prevents any clear message emerging from a glance. Hard to know what to suggest (put hourly station and model data in an appendix figure, so it's comparing like with like?), but I would at least encourage the authors to make the zero line more obvious.

*We have updated the figure with the zero line wider.*

Figure 8: A separate issue to the above, the spread (IQR etc) needs a clearer definition. Is it a spatial and temporal spread?

*In the case of figure 8 it is the temporal spread. We have clarified the caption.*

**Technical corrections (a full proofread is recommended)**

L45: "…part of the nitrogen oxides…" – nitrogen oxides include a lot more than NO and NO2 (e.g., N2O, N2O5 etc). To me this is also an example of curious wording. Suggest "Nitrogen dioxide (NO2; together with NO, a constituent of NOx) is a very well-established…"

*Fixed.*

L72: "The storm Ciara..." -> "Storm Ciara…" (and in other cases too). I'm no expert but seems like the preferred orthography is to capitalize the "S" in Storm when referring to a named one.

*Fixed*

L140: "A number of named extratropical cyclones (Storms Ciara, …)"

*Fixed*

L269: This is an example long paragraph that could be broken into shorter ones.

*The paragraph has been broken down into three.*

L326: Spell out TNO

*Done*

L399: This sentence doesn't make sense

*We have now clarified the sentence*

L429: This is not a stand-alone sentence (belongs as a clause of previous sentence).

*Fixed*

---

## Author Comment (AC2)

*We would like to thank the reviewer for their comments that helped to improve the paper's quality. Please read our answers in italic fonts below.*

This paper addresses the impact of the European 2020 Covid lockdown on NO2 levels using satellite data, surface NO2 data and model simulations. Because of the short TROPOMI satellite record the impact of meteorology is derived using a machine learning algorithm, which is also applied to the surface data. The large impact of meteorological differences between 2019 and 2020 is noted and this serves as a caveat to some previous simple presentations of the data during/after lockdown. The impact of lockdown is quantified for all large European cities by the 3 methods. These are important and useful results to publish, in a timely manner given the interest in the impact on the effects of lockdown. I only have minor comments and I think that the paper is publishable. My main comment is that the reader does not get a feel for the ML methodology and how well it works pictorially. Text refers to large outliers but it is important to show this to the reader (see my comment on Figure 5).

*We have added an additional figure providing times series for 2019 and 2020 for Madrid, showing the performance of the machine learning model. We also have amended the text to include the figure description and discussion.*

The paper is readable as is, but there is a very large number of minor grammatical errors which will need addressing. Maybe the ACP office will do that. I don't have time to go through them all, but I would point out that the typos start in affiliation 1 for the lead author (Forecasts not Forecast and Shinfield not Sinfield!). Not a good start. In fact the errors start in the paper title (which would need a hyphen: lockdown-induced).

*The grammar has been revised*

Other Specific Comments

Section 2.1 line 115. Give the local time of the TROPOMI observations in this section.

*Done*

Line 197. 'not expected'. Make it clear that this is not expected based on emissions. One could expect this if one understood the impact of meteorology.

*The sentence has been clarified.*

Line 225. 'Contrary to' change to 'In contrast to. . .'

Line 231. Table 1. Spell out the acronyms in the table headings.

*As requested by the other reviewers, this table has been removed from the article.*

Line 239-240. Explain what is meant by 'overfitting', what the implications would be and how you know it is not occurring.

*We have detailed the text accordingly.*

Figure 5 shows median values and not the mean. How different would Figures 3 and 4 be if the median was used? This needs some more explanation and somehow the same methodology should be included in one of the cases. You could make Figure 5 into 4 panels and show both methods. It is important to show the limitations of the ML method and provide the equivalent results to the other methods.

*For consistency we now have changed figure 3 and 4 with median estimates.*
*We provide here in the response the additional plots comparing the median vs mean on the satellite ML estimates. This provides a fairly similar overall picture. See below.*

[Figure]

*To prevent cluttering the paper and overwhelming the reader with lots of different statistics we prefer to keep the median estimates and not add the mean estimates. This is also justified by keeping consistency with the final figure that provide the equivalent of box plots based on the median and quartiles. Such estimates have also the advantage of being non-parametric displaying the variation in the statistical distributions without making underlying assumptions (i.e. gaussianity). We have now clarified the text accordingly.*

Line 266. 'would be expected'. How large is the interannual variability on NO2 emissions?

*We have clarified the text accordingly. It is true that the trend in NO2 reduction might influence the results as we only used 2019 to train the model.*

Line 288. 'Contrary to' -> 'in contrast'.

*Done*

Line 294. Same comment as above on overfitting.

*We now refer to the previous section in the text.*

Line 312. What does 'marginal' mean here? Small? Better to say what the lifetime of NO2 is and say that the impact is likely small.

*We are assuming that the reviewer refers to line 309 instead of 312. We have clarified the statement accordingly.*

Figure 6 caption. Say that these data are weather-normalized.

*Done*

---

## Author Response (AR2)

Dear Editors,

Please find the last corrected version of the manuscript. We have thoroughly checked and revised the grammar and as requested by the reviewers. We hope this version will be suitable for publication.

Kind regards